# Pulsed Electric Fields in Oncology: A Snapshot of Current Clinical Practices and Research Directions from the 4th World Congress of Electroporation

**DOI:** 10.3390/cancers15133340

**Published:** 2023-06-25

**Authors:** Luca G. Campana, Adil Daud, Francesco Lancellotti, Julio P. Arroyo, Rafael V. Davalos, Claudia Di Prata, Julie Gehl

**Affiliations:** 1Department of Surgery, Manchester University NHS Foundation Trust, Oxford Rd., Manchester M13 9WL, UK; francesco.lancellotti@mft.nhs.uk; 2Department of Medicine, University of California, 550 16 Street, San Francisco, CA 94158, USA; adil.daud@ucsf.edu; 3Department of Biomedical Engineering and Mechanics, Virginia Tech, Blacksburg, VA 24061, USA; jparroyo@vt.edu (J.P.A.); davalos@vt.edu (R.V.D.); 4Institute for Critical Technology and Applied Sciences, Virginia Tech, Blacksburg, VA 24061, USA; 5Department of Surgery, San Martino Hospital, 32100 Belluno, Italy; claudiadipratamd@gmail.com; 6Department of Clinical Oncology and Palliative Care, Zealand University Hospital, 4000 Roskilde, Denmark; kgeh@regionsjaelland.dk; 7Department of Clinical Medicine, Faculty of Health and Medical Sciences, University of Copenhagen, 1165 Copenhagen, Denmark

**Keywords:** cancer, electroporation, irreversible electroporation, gene electrotransfer, electrochemotherapy, tumour-treating fields, quality of life

## Abstract

**Simple Summary:**

Locoregional therapies play an ever-increasing role in contemporary oncology. This review provides an up-to-date, informed analysis of locoregional therapies harnessing electric pulses. Irreversible electroporation (IRE), gene electrotransfer (GET), electrochemotherapy (ECT), calcium electroporation (Ca-EP), and tumour-treating fields (TTF) are integral to the therapeutic strategy in several solid tumours, ranging from skin cancers to visceral and bone metastases. Still, despite consolidated credibility and a favourable trade-off between efficacy and side effects, these therapies face fragmentation, as shown by differences in the stage of development and regulatory approval worldwide. Here, leading experts convened at the 4th World Congress of Electroporation (Copenhagen, 9–13 October 2022) provide a coherent, updated snapshot of this field. Hopefully, these techniques’ common scientific and technological ground will allow researchers to overcome knowledge barriers and develop synergistic strategies to improve patient outcomes.

**Abstract:**

The 4th World Congress of Electroporation (Copenhagen, 9–13 October 2022) provided a unique opportunity to convene leading experts in pulsed electric fields (PEF). PEF-based therapies harness electric fields to produce therapeutically useful effects on cancers and represent a valuable option for a variety of patients. As such, irreversible electroporation (IRE), gene electrotransfer (GET), electrochemotherapy (ECT), calcium electroporation (Ca-EP), and tumour-treating fields (TTF) are on the rise. Still, their full therapeutic potential remains underappreciated, and the field faces fragmentation, as shown by parallel maturation and differences in the stages of development and regulatory approval worldwide. This narrative review provides a glimpse of PEF-based techniques, including key mechanisms, clinical indications, and advances in therapy; finally, it offers insights into current research directions. By highlighting a common ground, the authors aim to break silos, strengthen cross-functional collaboration, and pave the way to novel possibilities for intervention. Intriguingly, beyond their peculiar mechanism of action, PEF-based therapies share technical interconnections and multifaceted biological effects (e.g., vascular, immunological) worth exploiting in combinatorial strategies.

## 1. Introduction

The medical application of electricity has been documented since antiquity when eminent physicians such as Hippocrates, Scribonius, and Galen employed the electric discharges produced by torpedo fish to treat varied disease conditions [1]. However, it was only in the eighteenth and nineteenth centuries that a better understanding of the electricity phenomenon and the advent of the electric battery and other energy devices led to its systematic application (Figure 1).

Interestingly, the late twentieth century witnessed the deciphering of the biological effects of electricity and the development of novel therapeutic approaches to treat tumours based on the application of electric pulses. From the treatment of small, superficial cancers to that of parenchymal malignancies, pulsed electric fields (PEF) have gained a consolidated role in the treatment of solid tumours. Since their first successful application to introduce cytotoxic agents into cancer cells [2], in just a few decades, PEFs have entered the oncology armamentarium in the form of different approaches [3]. These therapies include irreversible electroporation (IRE), gene electrotransfer (GET), electrochemotherapy (ECT), calcium electroporation (Ca-EP), and tumour-treating fields (TTFs). Interestingly, they opened unprecedented therapeutic opportunities for patients affected by different types and stages of cancer while being associated with limited toxicity and a positive impact on quality of life (QoL).

Although the range of applications of PEF-based therapies is vast, encompassing a wide range of histologies and anatomical locations, they share a common ground. Among the varied biologic effects induced by PEF, cell permeabilisation (electroporation, [EP]) is the most relevant. Depending on the electric field strength, cell permeabilisation can be transient (reversible EP), thus allowing for the passage of nucleotides (DNA, RNA) [4], chemotherapy drugs (bleomycin, cisplatin) [5], or ions (e.g., calcium) [6,7], or permanent (irreversible EP, i.e., IRE), thus leading to loss of cell integrity [8,9] (Figure 2). TTFs stand out among PEF-based therapies as a type of electric field (low voltage, intermediate frequency), modality and duration of application (transdermal, through external contact electrodes applied for several hours a day), and mechanism of action (mainly, an antimitotic effect) (Figure 2). Nonetheless, preclinical studies have shown additional downstream effects of TTFs on cancer cells, including stimulation of autophagy, delay of DNA repair, induction of antitumour immunity, suppression of cancer cell migration and invasion, and also permeabilisation of the cell membrane and blood–brain barrier [10,11].

Current PEF-based therapies include five techniques based on different equipment, pulse protocols, and operating procedures. Despite being at different stages of development, they have shown efficacy in a wide range of tumours (Figure 3) and few side effects [12,13,14,15,16].

With high-voltage pulse electric fields, although it depends on the tissue type and treatment parameters (pulse number, frequency, duration), the electric field distribution determines the extent to which the treatment affects surrounding cell types. The electric field distribution is dictated by the electrode geometry and field applied. Traditionally for electroporation-based therapies, the field strength is highest near the electrodes. It dissipates outward with IRE occurring closest to the electrode, followed by an outer zone of reversible EP, and beyond this area, the weakened electric field results in no change to cell permeability [17]. The electrodes can be applied superficially, percutaneously, or laparoscopically. In TTFs, instead, alternating current electric fields are delivered using two orthogonal transducers applied to the skin over an extended time (indicatively, 18 h a day). The optimal frequency has been determined experimentally and varies by tumour type (for example, 200 kHz for brain cancer), whereas the reported electric field intensity ranges from 1–5 V/cm [10].

Another common feature across PEF-based treatments is that a considerable difference in sensitivity between normal and malignant tissue has been observed [7,18,19,20]. This difference in sensitivity is likely a combination of several factors. Thus, for example, it has been shown that malignant cells seal more slowly than normal cells [21], and if the membranes are permeable for a longer time, more drug molecules can be internalised into cells. Furthermore, tumour vasculature is more sensitive to the EP procedure and ECT [22], adding to differential toxicity. In addition, there are specific mechanisms for bleomycin, as used in ECT, that further augment differential effects, as there are specific mechanisms involved with calcium homeostasis that come into play in Ca-EP, thus benefitting the survival of normal cells. This selectivity aligns with clinical observations where the preferential targeting of tumour tissue is apparent [23]. For GET, a pattern of selective tumour targeting both at the local and systemic levels has also been reported [24].

Important and interesting questions remain, and work needs to be done to elucidate the underlying mechanisms of action and clarify the role of these therapies in each cancer algorithm. In addition, clinicians still lack a proper understanding of their principles, current applications, and ongoing research and opportunities. This gap is likely caused by the heterogeneity of the cancer teams involved; additionally, PEF-based therapies rely on different equipment and may require (as in the case of IRE and variable-geometry ECT [VG-ECT]) additional competencies such as interventional radiology or image-guided surgery. Lastly, a further impediment is their heterogeneous regulatory approval worldwide.

The 4th World Congress of EP (WCE, Copenhagen, 9–13 October 2022, https://wc2022.electroporation.net/, accessed on 25 October 2022) assembled more than 400 participants from 31 countries who discussed the up-to-date results and future research directions of PEFs in oncology and other research areas. This review aims to inform clinicians about the current applications and research directions on electric fields in oncology. As such, it provides an agile overview of the biological mechanism of each technique, presents the most consolidated clinical applications, and illustrates persisting challenges and cutting-edge research, including combination with novel immunotherapies. When considered in the broader context, PEF-based therapies certainly have a common ground at various levels (e.g., applied energy, instrumentation, biological mechanisms of action) on which we hope to overcome barriers, promote cross-contamination, expand their use, and develop novel therapeutic strategies.

## 2. Irreversible Electroporation

### 2.1. Principles of IRE

EP is a biophysical phenomenon in which nanoscale defects or “nanopores” are generated on the cell plasma membrane when exposed to an external electric field. By increasing cell transmembrane potential beyond a critical threshold of ~0.2–1 V, the lipid molecules along the phospholipid bilayer shift, creating temporary hydrophilic openings [4,25,26,27]. Transient membrane disruption (reversible EP) was initially designed for delivering chemotherapeutics [2] and gene therapies [4]; however, protracted or permanent nanopore formation leads to cell death. This phenomenon, called IRE, was previously an unwanted consequence of applying strong electric fields for biomedical applications. The cell death caused by IRE occurs without significant thermal heating or subsequent damage [9]. This initial theory was validated the following year through an in vivo rat study demonstrating IRE’s capability to produce a sizeable soft tissue ablation while effectively minimising thermal heating [28]. This ablation modality has been shown to spare surrounding nerve fibres [29] and extracellular matrices and minimise the heat sink effect [30], a phenomenon that hinders the effectiveness of other treatment modalities when the target lesion is in close proximity (<1 cm) to a large vessel (≥3 mm in diameter); the flowing blood, in fact, causes a cooling effect, which reduces the actual ablation volume.

Multiple in vivo studies have investigated the efficacy of IRE in the prostate [31], pancreas [32], liver [28,33,34], lung [35], kidney [36,37], breast [38], skin [39], heart [40], and brain [41]. Importantly, preclinical studies clarified important operational aspects, such as the need for neuromuscular blocking to reduce muscle contractions caused by electrical stimulation and ECG synchronisation to prevent the induction of cardiac arrhythmias. Clinically, IRE is delivered using needle electrodes inserted into the target lesion under ultrasound (US) or computed tomography (CT) guidance with the patient under general anaesthesia. Two to six monopolar needle electrodes are inserted depending on tumour size (generally between 1.0 and 5.0 cm) and anatomical location. Hence, the patient receives 60–100 high voltage (1.5–3 kV) bursts of 80–100 µs duration between the electrodes [42]. These pulses disrupt cell homeostasis, leading to necrosis, apoptosis, and autophagy within 8–24 h [43].

At the time of writing, there are more than sixty completed and ongoing clinical trials with IRE [44]. This section focuses on studies conducted over the last five years and includes some of the most recent clinical experiences, whereas Aycock et al. reviewed earlier studies [42]. From a practical standpoint, since the IRE application has high requirements for needle electrode insertion and parameter settings, a suboptimal procedure (e.g., incomplete coverage of the target volume) can affect the treatment outcome (e.g., incomplete response or local recurrence). Therefore, it is imperative to standardise the operative procedures, concentrate treatments at referral centres, and establish shared treatment indications/contraindications and management schedules within a multidisciplinary team.

### 2.2. Prostate Cancer

Radical prostatectomy or external beam radiotherapy is the standard of care in prostate cancer. However, these modalities are associated with side effects such as incontinence, urgency, and erectile dysfunction [45]. IRE can potentially reduce this risk by selectively targeting the tumour while sparing the surrounding tissues. Interestingly, a recent randomised trial with 106 patients has shown that focal (i.e., on the area of the prostate with positive biopsy cores) IRE ablation has similar short-term oncological outcomes and superior QoL to extended (i.e., zonal ablation) IRE [46].

Encouraging results also come from a mono-institutional study in Sydney with 229 patients with a median follow-up of 60 months following IRE. Participants were assessed at six months with magnetic resonance imaging (MRI) and at 12 months with transperineal biopsies. The six-month MRI showed eradication in 82% of patients, whereas residual clinically significant disease was found in 24% of patients in the one-year follow-up biopsy. Overall, 38 patients (17%) progressed to radical treatment at a median of 35 months after IRE [47].

Various experiences support the notion that patients treated with IRE for prostate cancer witness decreased prostate-specific antigen (PSA) levels and experience a better QoL [48,49,50]. In addition, rare posttreatment complications include urinary strictures [51,52], dysuria, haematuria, urinary tract infections, urgency [49,50,52], and erectile dysfunction [49,50,51,52]. Notably, complications commonly occurring with traditional procedures, such as urinary incontinence and erectile dysfunction, are drastically reduced when utilising IRE with a rate of 100% patient continence and 97% patient potency [53].

In addition to its use in primary prostate cancer, IRE has been used as a salvage treatment for local recurrences after radiotherapy [51]. A recently published mono-institutional study with 74 patients provides convincing data towards IRE’s short- to mid-term safety, oncological, and QoL outcomes in this setting. After a median follow-up of 48 months, local control was achieved in 57 patients (77%). Among patients who returned QoL questionnaires, 93% had preserved urinary continence, and 23% had sustained erectile function. Complications included one rectal fistula, whereas urethral sloughing occurred in eight cases and required transurethral resection [54]. This complication is likely due to unintentional tissue heating when rapidly delivering an electric current to previously irradiated fibrotic tissue. Similar results have been shown in the two-centre prospective focal IRE (FIRE) trial [55].

Interestingly, various imaging modalities have been utilised to assess tumour response, such as multiparametric MRI (mpMRI) [56], contrast-enhanced ultrasound (CEUS) [57], and prostate-specific membrane antigen (PSMA) positron emission tomography—CT (PET-CT) [58]. Nevertheless, tissue biopsy remains the gold standard for accurately evaluating tumour response or recurrence. Finally, imaging modalities used in conjunction with IRE, such as the MRI-transrectal US fusion-guided IRE, provide novel opportunities to increase the accuracy of treatment delivery [59].

### 2.3. Pancreatic Cancer

IRE’s role in the treatment of pancreatic ductal adenocarcinoma (PDAC) is versatile [60]. Unfortunately, only a minority of these patients are candidates for surgical resection due to locally advanced disease (locally advanced pancreatic cancer [LAPC]). In this highly challenging scenario, IRE may be an option. Most published literature on IRE in PDAC is on stage III/IV LAPC, whereby the role is palliative, including local tumour control and pain management [61,62,63,64].

Additionally, and interestingly, IRE can be used as a consolidative treatment following chemotherapy or chemoradiotherapy; alternatively, it can be an adjunct to surgery in patients with borderline resectable disease [65]; finally, it can be used in the control of local recurrence after previous curative surgery or palliation in patients unfit for surgery or chemotherapy [66,67]. A recent clinical trial from Eastern Asia involving 74 patients treated with induction chemotherapy and IRE during open surgery at 11 centres reported a 5-year progression-free survival (PFS) rate and a 5-year overall survival (OS) rate of 28.8% and 31.2%, respectively. The authors observed 30 complications that occurred in 13 (17.6%) patients, which were primarily associated with the direction of electrode placement and gastrointestinal disease infiltration. In multivariate analysis, the type of chemotherapy regimen was the only significant factor associated with PFS and OS [68]. Other studies investigated the efficacy of IRE and chemotherapy versus standard-of-care treatments such as chemotherapy alone [69] or the combination of induction chemotherapy and radiation [65] in patients with LAPC. These trials indicated improved PFS and OS for the combination of IRE and chemotherapy. Interestingly, the PANFIRE phase II trial aimed to deliver IRE percutaneously, thus avoiding the necessity for laparotomic surgery. Median OS was 17 months, regardless of induction chemotherapy [70].

Concerning the modality of IRE application, it is worth noting that clinicians have utilised this technique to treat surgical resection margins to target residual microscopic disease [71]. Thus, IRE is an attractive tool for the so-called “margin accentuation” (MA) near the superior mesenteric artery, celiac trunk, superior mesenteric vein, and portal vein to achieve lower rates of positive margins. This approach is generally directed towards three critical anatomical areas, i.e., pancreas neck margin, superior mesenteric/portal vein margin, and superior mesenteric artery margin [12]. The initial use of IRE for MA (MA-IRE) was published by Kwon et al. using two IRE electrodes in bracketing and parallel to the vascular structure (artery or vein, or both) to enhance the resection margin [62]. There are few reports of intraoperative IRE and margin accentuation followed by pancreaticoduodenectomy in borderline resectable pancreatic cancer (BRPC), suggesting a significant role in reducing the risk of local recurrence and improving survival [61,72]. Martin et al. enrolled 200 patients with LAPC who underwent IRE alone (*n* = 150) or surgery plus IRE on resection margins (*n* = 50). All patients received induction chemotherapy, and 52% received chemoradiation therapy before IRE. Thirty-seven per cent of patients suffered from complications, with a median grade of 2 (range, 1–5). The median length of hospital stay was six days. With a median follow-up of 29 months, six patients (3%) experienced local recurrence; of note, the median OS was 28.3 months in the surgery-IRE group and 23.2 months in the IRE group. These results support that adding IRE to conventional chemotherapy and radiation therapy prolongs survival compared with historical controls [61]. Martin et al. investigated the rates of local recurrence and margin positivity in patients with BRPC who underwent pancreatectomy with (*n* = 75) or without (*n* = 71) IRE-MA using a prospective database. This study showed that IRE-MA can be performed safely and effectively. In particular, local recurrence rates and disease-free intervals were similar between groups. Finally, the IRE-MA group’s OS was significantly higher than the pancreatectomy-alone group [73]. Kundalia et al. evaluated the margins positivity rate of IRE-MA in pancreatic head tumours and compared disease-free survival (DFS) and OS with a retrospective control group. They reported a trend towards reduced margin positivity (from 51.6% in the control group to 35.0% in the IRE-MA cohort) and no significant differences in OS and DFS.

Notably, using IRE for MA is not associated with increased postoperative complications [12]. Numerous studies have reported on the safety and feasibility of IRE [68,70,74]. The main treatment-related adverse events include complications to the pancreas (pancreatitis, fistula, abscess), liver (bile leak, biliary peritonitis, cholangitis, abscess), bowel (ileus, perforation, bleeding, fistula), vessels (pseudo-aneurysm, hepatic arterial thrombosis, non-occlusive superior mesenteric vein/portal vein thrombosis), and spleen (infarction). Although rare, patients may suffer from multiple types of complications. In the largest series, all-grade morbidity is in the range of 36–40% [61,73]. Therefore, weighing trade-offs between benefits and harms is imperative. To date, the optimal timing and patient selection criteria for IRE remain debated. The recent analysis of a multicentre database with 187 LAPC patients treated with induction chemotherapy followed by open IRE has suggested patient age, serum CA 19-9, no previous irradiation, and induction chemotherapy with FOLFIRINOX-gemcitabine/abraxane as possible selection criteria [75].

### 2.4. Liver Malignancies

Managing hepatic malignancies, even when localised, is challenging due to the frequent association with parenchymal disease and complex anatomy [76]. By sparing the structural framework of adjacent vessels and tissues while inducing apoptosis in the targeted cells, IRE allows treating tumours adjacent to critical structures whose critical location represents an exclusion criterion from other ablative treatments [77]. IRE is a valuable option for tumours near the hepatic artery and hepatic hilum in this setting. A recent analysis by Gupta et al., including 25 studies in patients with hepatocellular carcinoma (HCC), intrahepatic cholangiocarcinoma (ICC), or colorectal cancer liver metastasis (CRLM), indicates a collective 3-year PFS of 49% and 3-year OS of 41%, with better outcomes in the HCC subgroup [78].

#### 2.4.1. Primary Liver Cancers

According to a single-centre retrospective analysis, IRE has efficacy in treating early-stage HCC not amenable to standard ablative techniques, with an excellent complete response (CR) rate and long-term local control, particularly in small lesions. In this study, 23 patients received IRE for 33 HCC with a median 2.0 cm tumour size. Twenty-nine (87.9%) tumours were ablated after one (*n* = 26) or two (*n* = 3) procedures. The median local-recurrence-free survival was 34.5 months [79].

#### 2.4.2. Liver Metastases

IRE’s ability to specifically treat CRLM was demonstrated in COLDFIRE-1, an ablate-and-resect study [80], and COLDFIRE-2 [81], a phase II open-label clinical trial. As a result, the latest multidisciplinary consensus guidelines recommend IRE use for perihilar or perivascular CRLM [82]. In particular, IRE is recommended for 3–5 cm tumours when additional systemic therapy is not viable. The safety of IRE within 1.0 cm from critical structures is evident. Additionally, the procedure is efficacious even for tumours abutting or encasing large high-flow vascular structures such as the portal vein and hepatic arteries [83,84]. Other typical applications of IRE are tumour ablation near bile ducts, the gallbladder, or the bowel [85,86,87]. The evidence suggests superior results in lesions ≤3 cm in size [79,84]. Postoperative complication rates after IRE are comparable to that of radiofrequency ablation (RFA) and microwave ablation (MWA) [88]. The most common major complication is a liver abscess, whose incidence correlates with the presence of bilioenteric anastomosis [89]; subcapsular hematoma or need for laparotomy due to haemorrhage, arterioportal fistula, bile leak, post-procedure biliary strictures, and post-IRE liver failure occur rarely. As IRE is typically reserved for patients who are not RFA or MWA candidates, a direct, matched, and unbiased comparison between IRE and thermal ablation is not feasible. Additionally, a direct comparison between IRE, surgery, and stereotactic body radiation therapy is impossible.

Additionally, most participants in IRE clinical trials have been heavily pre-treated; therefore, comparing overall survival rates with other techniques carries an inherent bias. Finally, as patients underwent several other anti-cancer interventions before and after IRE, no PFS or OS outcomes can be attributed solely to this intervention. In the COLDFIRE-2 trial, 51 patients with CRLM up to 5 cm underwent IRE during either an open or percutaneous procedure. The per-patient 1-year local PFS was 68%. Following repeat procedures, local control was achieved in 74% of participants. A total of 23 patients experienced 34 adverse events (complication rate, 40%) [81]. A range of complications can occur when delivering IRE to the liver. These vary from minor side effects such as mild pain and fever [90] to major complications such as hydrothorax [90], pneumothorax, and brachial plexus injuries [91]. The overall complication rate is 23.7% from collective data, with severe adverse events arising in 6.9% of cases [78].

### 2.5. Localised Renal Cell Carcinoma

While the standard of care for localised renal cell carcinoma is partial nephrectomy, professional societies increasingly accept alternative strategies thanks to low complication rates and comparable oncologic outcomes [92]. Percutaneous ablation is particularly attractive in patients with significant comorbidities, renal impairment, old age, recurrent and multiple hereditary renal cell carcinomas or those unwilling to undergo surgery [93]. Although the experience with IRE is limited compared with traditional ablative modalities, this technique has the potential to overcome limitations of thermal ablation, enabling the ablation of small renal masses near vital structures. The procedure has proved feasible and safe [94], although short- and mid-term oncological outcomes appear inferior to other treatments due to residual microscopic disease leading to local recurrence [95]. In fact, according to a 2017 review of 41 patients, the 2-year local-recurrence-free survival was 83% [96], and another retrospective analysis of 47 patients published in 2019 reported a 5-year local-recurrence-free survival of 81.4% [97]. The early operator’s learning curve may partially explain suboptimal results, and more recent series indicate improved oncological durability [98].

### 2.6. Research Directions—IRE

#### 2.6.1. Gastrointestinal Tract

Upper gastrointestinal tract malignancies have only recently been treated with IRE. In this setting, specifically designed catheter electrodes have been designed that are suitable for coupling with an endoscope to allow visualisation and avoid hollow viscus perforation. A recent study used a finite element analysis of the digestive tract to simulate the effects of electric fields and thermal dispersions emitted from different catheter configurations to assist with treatment planning [99]. Furthermore, Jeon et al. investigated catheter-based IRE within in vivo pig models to identify upper threshold voltage limits in the oesophagus, stomach, and duodenum [100]. This study revealed potential complications from intense electrical fields in these tissues, such as perforation and bleeding, as the critical voltage required the production of an ablative effect. Despite the launch in 2015 of a clinical trial on IRE in patients with unresectable oesophageal cancer, no results have been reported so far [101].

#### 2.6.2. Immune Effect and Combined Strategies

Beyond ablation, IRE induces an immune response. The initial cell death is followed by the release of damage-associated molecular patterns (DAMPS), which stimulate the antigen-presenting cells (APC). These, in turn, migrate to the regional lymph nodes, where they stimulate an antitumour response by priming T cells [3,102]. This effect of IRE—called immunogenic cell death (ICD), has the potential to elicit a systemic immune response and produce the abscopal effect, where off-target tumours are recognised and targeted by the adaptive immune response [103]. In this regard, IRE, similar to other local therapies [104], acts as an “in situ” vaccination to elicit a systemic immune response [105,106]. Several studies have investigated the immunomodulatory effect of IRE. Interestingly, IRE has been shown to eliminate PD-L1-positive tumour cells, which have an inhibitory effect on PD-1-positive lymphocytes [71]. Furthermore, a sharp decrease in regulatory T cells (Tregs), which inhibit cytotoxic T cells and dendritic cells, has been shown [107,108]. Taken together, these effects promote immune cell infiltration and the activation of the antitumour immune response.

Conversely, there is evidence that the increase in IFN-γ produced by the immune infiltrate may increase PD-L1 expression in the residual tumour cells [109]. This highlights the importance of achieving complete tumour ablation and associating other (immune)therapies to achieve an effective immune response. Tracking immune cell subpopulations in the tumour microenvironment following IRE has opened exciting opportunities for the association with immunotherapy. For example, two clinical studies using patients with LAPC investigated PD-1 immune checkpoint inhibition to prevent the remaining tumour cells from suppressing the antitumour response [109,110]. Of note, anti-PD-1 immunotherapy was associated with a significant increase in OS and PFS compared with IRE alone (44 vs. 23 months and 27 months vs. 10 months, respectively [110]. In another clinical study, researchers administered allogenic, cytotoxic immune cells (i.e., Vγ9Vδ2 T cells) [111] or allogenic natural killer (NK) cells [112] in combination with IRE. In murine models, stimulator of interferon genes (STING) agonists have shown efficacy as an adjuvant due to their ability to increase IFN-γ production from the tumour, which, together with DAMPs, stimulates APC maturation [113,114]. While, encouragingly, this effect may promote an abscopal effect, on the other side, it has been observed that post-IRE residual tumour cells may be more resistant due to the increased presence of IFN-γ. Finally, the use of multiple immunotherapies to assist in the various steps of the antitumour response has proven efficacious in the preclinical setting. For example, a study used a TLR7 agonist to stimulate the innate immune system in a murine model along with IRE and anti-PD-1 agents to generate a sustained immune response [105]. Similarly, Peng et al. demonstrated that combining IRE, anti-PD-1, and TGF-β inhibitors was more effective than any of them alone [115].

#### 2.6.3. High-Frequency Irreversible Electroporation

Introduced in 2011 by Arena et al. [116], a second generation of IRE was developed to reduce electrochemical effects, enhance cell selectivity toward malignant cell types [117], mitigate nerve and muscle excitation, and negate the need for cardiac synchronisation [118] by delivering bursts of short, bipolar pulses (0.5–10 µs) into heterogeneous tissue. This new technology was coined ‘high-frequency irreversible electroporation’ (H-FIRE). Since its emergence, H-FIRE has been validated to treat malignancies in vivo within the breast [119], liver [120,121], and brain [122]. In rodent models, H-FIRE has been shown to cause significant blood–brain barrier disruption [123,124] and immune infiltration [119,122] when targeting various malignant, promising avenues for future combinatorial therapeutics with this ablation modality. H-FIRE has also shown efficacy in selectively treating myocardial tissue as a remedy for cardiac arrhythmias [125,126]. In 2018, an initial H-FIRE clinical study treated 40 prostate cancer patients without ECG synchronisation. Following six months post-treatment, all 40 patients sustained urinary function, and 14 of 14 patients maintained sexual potency [127]. A recent expansive clinical evaluation by Wang et al. [128] saw 109 prostate cancer patients pooled from four centres treated with H-FIRE and evaluated after six months. Of 100 patients who underwent a biopsy, 14 developed recurrences in the prostate, with six patients classified as having clinically significant prostate cancer (csPCa) (5 outside and one inside of the treatment zone). Compared with similar ablation modalities, the rate of residual csPCa was significantly lower, making H-FIRE a rising, competitive clinical therapeutic. Lastly, 98% of patients retained urinary function, with 9% of patients experiencing emergent sexual dysfunction at six months.

## 3. Gene Electrotherapy

### 3.1. Principles of GET

Neumann et al. initially described the application of short electric pulses to deliver macromolecules to mouse cells in 1982 using a chamber with opposed electrodes where dissociated cells were suspended in a conducting liquid [4]. Several pioneering investigators expanded and modified this concept in the next three decades by using electrodes to deliver macromolecules in vivo. A striking observation from these studies was that plasmid DNA could deliver immune active cytokines such as GM-CSF, IL-2, or IL-12 into syngeneic tumour models in mice and could produce local regression and a “vaccine” effect [129]. In vivo EP with IL-12 was tested in a preclinical toxicity study in the B16-F10 model and, similar to other observations, proved to be well tolerated without any significant organ toxicity [130]. In these preclinical experiments, schedule and dose, as well as the cytokine used, appeared to influence tumour control, and repeated treatment at short intervals was more effective than single treatments, and IL-12 stood out for its effectiveness [131]. Based on these experiments, a phase I trial was launched in 2006 to explore IL-12 plasmid EP in patients with melanoma and solid malignancies [24]. In this trial, patients were treated with pIL-12 EP on accessible tumours on days 1, 5, and 8 with escalating plasmid doses, and only one treatment cycle was delivered. The treatment was well tolerated, and no maximum tolerated dose was identified. Responses were observed at several doses, and two patients had a CR in all tumours, including those uninjected. Correlative studies indicated that IL-12 protein and IFN-γ, a downstream product of IL-12, peaked by day 11 and gradually decreased over 2–4 weeks. No systemic spillage of IL-12 was observed, explaining the lack of systemic toxicity. Based on the phase I trial results, phase II trials were conducted in melanoma [13,132], Merkel cell carcinoma [133], breast cancer [134], and cutaneous T-cell lymphoma (NCT01579318).

### 3.2. Melanoma

In melanoma, the treatment showed an objective response rate (ORR) of 35.7%, with 17.9% of patients having a CR. The median OS was in excess of 29.7 months, and 46% of patients had a regression in at least one uninjected lesion. While increased immune activation was seen in both treated and untreated lesions, there was also adaptive immune resistance, and perhaps for this reason, many patients with an initial response developed progressive disease. Interestingly, patients with progressive disease could, in many cases, be successfully treated with PD-1 blockade [13,132].

### 3.3. Merkel Cell Carcinoma

In Merkel cell carcinoma, the ORR was 25% (3/12), with two patients experiencing durable clinical benefit (16 and 55+ months, respectively) [133]. Interestingly, Bhatia et al. showed the expansion of Merkel Polyomavirus T antigen-positive T cells following IL-12 EP in this study.

### 3.4. Breast Cancer

Triple-negative breast cancer is an aggressive disease with limited therapeutic options. Antibodies targeting PD-1)/PD-L1 have entered the therapeutic landscape, but only a minority of patients can benefit from them. Using mouse models of triple-negative breast cancer, researchers evaluated immune activation and tumour targeting of intratumoural IL-12 plasmid followed by EP. Single-cell RNA sequencing of murine tumours identified a posttreatment activated gene signature associated with enhanced antigen presentation, T-cell infiltration and expansion, and PD-1/PD-L1 expression. Next, and interestingly, the assessment of pre-/post-treatment biopsies from patients from a single-arm prospective clinical trial confirmed the enrichment of this signature from patients that exhibited an enhancement of CD8+ T-cell infiltration following treatment [134].

### 3.5. Urology

Among urological cancers, GET with several plasmids encoding prostate cancer antigens has been investigated in early-phase clinical trials [135].

#### 3.5.1. Prostate Cancer

In a clinical phase I/II dose escalation trial on patients with prostate cancer, the plasmid encoding PSMA epitopes was transfected by EP of the muscle tissue. The response to the therapy was remarkably higher among patients who received DNA injection combined with EP compared with the control group of DNA injection alone [136]. Another phase I studied evaluated the feasibility of a prostate surface antigen (PSA) vaccine combined with intradermal EP. Before the vaccination, androgen deprivation therapy was administered to induce T-cell infiltration. GET successfully increased PSA-specific CD8+ T-cell-mediated immune response [137].

#### 3.5.2. Bladder and Renal Cancer

Preclinical studies in bladder cancer investigating the association of recombinant bacillus Calmette-Guerin (BCG) with IL-12 confirm that DNA electrotransfer increased INF-γ and IL-12 secretion as well as CD4+/CD8+ T cell and NK cell infiltration [135]. Similarly, in renal cancer, the tumour necrosis factor-related apoptosis-inducing ligand (TRAIL) or IL-12 cDNA plasmid electrotransfer successfully inhibited tumour growth in murine models [135].

### 3.6. Research Directions—GET

#### 3.6.1. Combination with PD-1 Blockade in Melanoma

Given the above data, several trials have been conducted and are ongoing in melanoma with pIL-12 EP. In a combination trial, PD-1 blockade with pembrolizumab was added to pIL-12 EP in patients with immunologically “cold” or quiescent tumours. These tumours were biopsied before treatment, and T cell content was determined using a flow cytometry assay [138]. Only those patients with a CD8 + PD-1 + CTLA4+ frequency below 25% were enrolled (these patients would not be expected to respond to monotherapy PD-1). The objective response rate for the combination of pIL-12 and pembrolizumab (the PD-1 antibody used) was 41%, and 36% of all treated patients had a CR in all tumours. Correlative analyses showed that the combination enhanced immune infiltration and sustained the IL-12/IFNγ feedback loop, driving intratumoural dendritic cells, which recruited T cells and resulted in a systemic immune response. In addition, the combination enhanced immune infiltration and sustained the IL-12/IFNγ feed-forward cycle, driving intratumoural cross-presenting dendritic cell subsets with increased TILs, emerging T cell receptor clones, and, ultimately, systemic cellular immune responses [132]. Based on these premises, a phase II trial on adding pIL-12 intratumoural EP to pembrolizumab was started in PD-1 refractory patients with confirmed disease progression. Preliminary data on the first 56 treated out of 100 planned patients indicate deep, durable responses in locally treated and distant visceral metastases without additional toxicity concerns. Of note, ORR was 30%, and tumour reduction was observed in untreated lesions in 12 out of 12 patients with inaccessible or accessible untreated lesions [139].

#### 3.6.2. Predictive Biomarkers

Several emerging themes can be discerned based on these preclinical and clinical data. A major question is the development of biomarkers to predict response to pIL-12 EP. While changes in the tumour immune microenvironment are seen with pIL-12 EP, pre-treatment biomarkers predicting response will be critical as more checkpoint inhibitors and other therapies enter the clinic and are explored in clinical trials. If patients with a specific biomarker can be identified ahead of treatment, those patients can be enriched in clinical trials and potentially be selected for treatment clinically.

#### 3.6.3. Cytokine/Chemokine Combinations

Another significant issue is combination cytokine or cytokine/chemokine combinations. Preclinical studies have identified CXCL9 as a promising chemokine for pIL-12 [140]. Other cytokines and antibodies could also be combined, including checkpoint and agonist antibodies, to avoid systemic toxicity, which can be considerable.

## 4. Electrochemotherapy

### 4.1. Principles of ECT

ECT combines rEP with the injection or infusion of a cytotoxic drug. Contrary to IRE, in ECT, the electric pulses are not intended to kill the cells at all but rather to transiently induce membrane permeability, thus allowing for the passage of drugs [141]. As a result, non-permeant or low-permeant agents such as bleomycin and cisplatin can diffuse and accumulate into tumour cells with an exponential increase in their cytotoxicity [141]. Importantly, ECT kills tumour cells selectively without harming normal surrounding tissue. Cell death is caused predominantly by drug-induced apoptosis [142,143,144], although a transient vascular lock, a composite antivascular action [145,146], and an immune response also occur [147,148].

Initially, ECT gained traction, mainly in Europe, as a local treatment for patients with small-size cutaneous and subcutaneous tumours, generally under local anaesthesia or mild general sedation. More recently, its application has expanded to the treatment of bone and intra-abdominal malignancies [149]. Since the initial experiments and seminal clinical experiences, the following parameters have been adopted by the European Standard Operating Procedures of ECT (ESOPE) [141,150]: eight consecutive square-wave electric pulses of 100-μs duration delivered at a repetition frequency of 1–5000 Hz; the voltages applied depend on the pulse applicator and the distance between the electrodes, and they are automatically set by the pulse generator.

The tenets of ECT are that the tumour must be simultaneously exposed to chemotherapy and covered entirely by electric fields of appropriate amplitude as a prerequisite to success. Therefore, the application of electric pulses is time-dependent following the chemotherapy administration into the bloodstream during the procedure. Specifically, the time window must be 1–10 min after intratumoural injection (concentration, 1000 IU/mL) and up to 40 min after intravenous infusion (concentration, 15,000 IU/m^2^) [150]. This means several applications are required to achieve proper tumour coverage. On a practical note, parallel plate electrodes are used for superficial tumours, whereas needle electrode arrays are used for deep lesions.

Furthermore, the selected drug can be delivered intravenously (bleomycin) or intratumourally (bleomycin or cisplatin). Finally, it is worth noting that ECT is classified according to the adopted pulse applicator and electrodes. As such, the following treatment modalities are available: standard ECT (also defined as fixed-geometry ECT [FG-ECT]), which employs short (maximum 5 cm in length) fixed-geometry needle or plate electrodes; variable-geometry ECT (VG-ECT), which employs long, freely placeable needle electrodes; and finally, endoscopic ECT, which uses deployable devices capable of reaching deep lesions with needle electrodes in narrow anatomical locations [14] (Figure 4). FG-ECT is time-tested and has entered the therapeutic algorithm of several superficial cancers [14]; VG-ECT has been approved in Italy for treating bone metastases [151].

### 4.2. Skin Cancers

#### 4.2.1. Melanoma

Since the introduction of ECT in clinical practice, superficially metastatic melanoma has represented one of the most frequent treatment indications. Skin metastases are a significant cause of morbidity in patients with advanced disease. Due to melanoma’s propensity for multifocal spread, ECT has proved a valuable option to achieve tumour control rapidly while avoiding the risks and toxicity of more invasive treatments such as isolated limb perfusion (ILP) or isolated limb infusion (ILI). According to recent analyses, including 27 studies and 1161 patients, ECT is associated with a 77.6% ORR and a 48% CRR, irrespective of the route of chemotherapy administration; moreover, the 1- and 2-year local control rate is 54–89% and 72–74%, respectively [152]. Notably, there is intriguing preliminary evidence that ECT enhances local anti-melanoma immunity by producing dendritic cell activation, decreasing CD4+ FOXP3+ T regulatory cells, and increasing CD3+ CD8+ T cells’ infiltration [147,148]. Recently, with the advent of checkpoint inhibitors, the landscape of melanoma treatment has dramatically changed, and the interest in local and locoregional therapies has shifted to combined therapeutic strategies. In this regard, a retrospective multicentre analysis from the InspECT group and the Slovenian Cancer Registry has shown better outcomes in patients who received pembrolizumab and ECT compared with pembrolizumab alone. In particular, the local ORR was 78% vs. 39%, and the 1-year local PFS was 86% vs. 51%, respectively. Additionally, and somewhat unexpectedly, although with the limitations of a retrospective analysis, 1-year systemic PFS was 64% vs. 39%, and 1-year OS was 88% vs. 64%, thus suggesting a beneficial systemic effect [153].

A recent significant contribution relates to the clinical benefit of ECT on patient QoL. The InspECT register collected prospective data from 378 melanoma subjects treated with ECT and assessed employing the EuroQoL (EQ-5D-3L) questionnaire at baseline, one, two, four, and ten months. EQ-5D and EQ-VAS scores remained within the minimal important difference (MID) boundaries following treatment, particularly among complete responders. Still, a subanalysis of the EQ-5D items revealed a transient deterioration in pain/discomfort and mobility domains and a persistent deterioration in self-care and usual activities. Conversely, concomitant checkpoint inhibition correlated with better EQ-5D and EQ-VAS trajectories. Finally, and interestingly, baseline EQ-5D was the exclusive independent predictor for CR, thus suggesting the introduction of QoL assessment as a screening tool to identify melanoma patients most likely to benefit from ECT [154]. 

Finally, ECT is applied with function-sparing intent in patients with melanoma of the perianal region and mucosal melanoma of the anal canal. This indication is a niche setting, and the clinical experience is limited to a few centres; nonetheless, preliminary results indicate the procedure’s feasibility, safety, and efficacy to ensure durable local control while preserving organ function [155,156,157].

#### 4.2.2. BCC

As the most common skin cancer, it is unsurprising that basal cell carcinoma (BCC) represents another typical ECT indication. In general, surgical resection is the mainstay of treatment for patients with small/intermediate-sized lesions in low-risk anatomical areas, whereas Mohs surgery is reserved for large or recurrent BCCs, particularly in high-risk anatomical areas. However, contemporary population ageing poses new therapeutic challenges, and patients with relevant comorbidities are best served by nonsurgical approaches, including radiation, photodynamic therapy, cryotherapy, topical agents, and ECT. Nonetheless, due to the large number of therapeutic options and the paucity of comparative trials, it is impossible to provide strong indications. Regarding ECT, a recently published prospective registry-based study aimed to describe ECT modalities and evaluate its efficacy, safety, and predictive factors in 330 patients (85% with primary BCC, 80% located in the head and neck, median tumour size, 13 mm) treated between 2008 and 2019. The procedure was carried out under local anaesthesia in 68% of cases, with the adjunct of mild sedation in the remaining 32%. Of 300 evaluable patients, 242 (81%) achieved a CR after a single application. Concerning predictive factors, treatment naïvety and intraoperative coverage of deep tumour margins with electrodes predicted CR achievement, whereas previous radiation showed an unfavourable correlation. At 17 months, 28 (9.3%) patients experienced local recurrence or progression. Despite no convincing evidence compared with standard surgical excision, ECT can still be considered an opportunity to avoid the morbidity of surgical resection in selected patients. To this aim, treatment naïvety, no previous radiation, and the ability to cover tumour margins may inform patient choice [158].

#### 4.2.3. Cutaneous Squamous Cell Carcinoma

Cutaneous squamous cell carcinoma (cSCC) is an aggressive skin cancer associated with a high risk of microscopic infiltration and recurrence rates. According to a recent analysis of the InspECT registry, including 162 individuals with primary or recurrent disease, the ORR was 83%, with complete remission in 62% of cases. Additionally, in a multivariate model, intravenous bleomycin and small (<3 cm) tumour size correlated with response and the patients with primary tumours achieved better local tumour control than those with recurrent locally advanced disease [159].

### 4.3. Skin metastases from Breast Cancer

Breast cancer is the most common solid tumour spreading to the skin, representing 24–50% of patients with superficial metastases. Clinical presentation may be heterogeneous, but the most frequent pattern is widespread chest wall involvement, which is generally not amenable to surgical treatment [160]. Additionally, there is no standard-of-care approach, and, despite a minority of resectable patients, the therapeutic strategy combines systemic treatment with various local therapies (radiation, ECT, photodynamic therapy, and topical agents) to ensure local control [161]. According to the most recent summary data, ECT is associated with 75% ORR and 46% CRR in breast cancer [162]. ECT has shown satisfactory activity and sustained local control in patients with refractory chest wall recurrence, particularly those with fewer and less scattered skin metastases. Partial responders and new lesions can be handled with additional ECT applications, albeit at the cost of increasing pain and toxicity [163].

### 4.4. Bone and Soft Tissue Tumours

Bone and soft tissue neoplasms represent a less common indication due to the technical complexity of the procedure in the case of deep-seated targets and, in the case of soft tissue tumours, their relatively lower incidence compared with skin cancer and cutaneous metastases.

#### 4.4.1. Bone Metastases

Bone metastases represent a challenging application setting because they cause pain, fractures, neural compression, and impaired mobility. Additionally, treatment delivery requires a dedicated pulse generator, long, freely placeable electrodes, and radiological guidance. Nevertheless, according to the updated experience of the Rizzoli Institute in 38 patients, ECT produced a 29% ORR and disease stabilisation in 59% of cases, with a significant decrease in pain compared with the baseline [164].

#### 4.4.2. Kaposi Sarcoma

Kaposi sarcoma is a relatively indolent vascular tumour with a predilection for a multifocal spread in the skin and subcutaneous tissue. Thus, it is not surprising that, in the few available studies, the CR rate ranges from 61% to 89% after a single course of treatment, with few side effects [165,166,167].

#### 4.4.3. Superficial Angiosarcoma

ECT has been applied with promising results in breast cancer patients who developed superficial angiosarcoma on lymphedema or following radiotherapy and in elderly patients with angiosarcoma of the scalp. Currently, only limited studies exist, and they have enrolled small populations. According to a couple of multicentre studies with 19 and 20 patients, the ORR was 63–80% and CRR 40–42% [168,169]. However, despite these encouraging results, soft tissue angiosarcomas exhibit distinct and challenging presentations, particularly in breast cancer patients, and require an aggressive, coordinated multidisciplinary approach [170,171].

#### 4.4.4. Deep-Seated Soft Tissue Sarcomas

The development of a novel pulse generator has allowed the application of electric pulses through the insertion of long, independent needle electrodes, which are longer than those commonly used to treat skin cancers with standard ECT. This equipment enables targeting up-to-20-cm-deep lesions and tailoring the electric field according to the tumour size and geometry, thanks to the support of radiologic image guidance [149]. Two mono-institutional studies have confirmed the feasibility and safety of VG-ECT in patients with soft tissue tumours so far [172,173].

#### 4.4.5. Vascular Malformations

Intralesional injection of a sclerosing agent is a standard treatment for vascular malformations, but multiple treatments are often required to achieve a response. According to the results of a recent prospective observational study on bleomycin electrosclerotherapy (BEST) in 30 patients with vascular malformations (predominantly venous malformations), CR or significant improvement was observed in 17 (57%) and 7 (23%) patients, respectively. Most of the patients were very satisfied with the treatment outcome. The most commonly reported complications were swelling, pain, and bleeding. Electrosclerotherapy is a promising method of augmenting the efficacy of intralesional bleomycin injections when treating vascular malformations. Specifically, it can reduce the administered dose and the number of treatment sessions [174]. Dedicated operating procedures are in preparation by a dedicated panel of disease experts within the InspECT network.

### 4.5. Intra-Abdominal Tumours

ECT can be applied using an open or laparoscopic approach in the interventional radiology suite or the operating theatre.

#### 4.5.1. Pancreas

ECT in pancreatic adenocarcinoma is not established. Small monocentric experiences suggest the procedure’s feasibility during open surgery in patients with locally advanced disease and the absence of serious adverse events [175,176].

#### 4.5.2. Liver

The experience with ECT in primary and secondary liver malignancies has been increasing steadily. The tumours treated so far include hepatocellular carcinoma (HCC), cholangiocarcinoma, colorectal liver metastases (CLRM), and breast cancer metastases. ECT has been applied intraoperatively (employing fixed- or variable-geometry electrodes) or percutaneously (using variable-geometry electrodes). According to the most recent cumulative data, the CR rate ranges from 33% to 100%, depending on tumour size [177].

### 4.6. Gastrointestinal Cancers

#### 4.6.1. Oesophageal Cancer

Endoscopic ECT is still in its infancy. In the first-in-human trial, published in 2018, six patients with advanced oesophageal cancer were treated with an appositely developed endoscopic pulse applicator and intravenous bleomycin. The procedure lasted 30–60 min and was carried out with no safety concerns; the main adverse events included retrosternal pain, fever, and pneumonia. On a technical note, tumour coverage was partial in three out of six patients due to impassable cancer stenosis [178].

#### 4.6.2. Colorectal Cancer

The first-in-human trial, published in 2020, enrolled seven patients with primary or recurrent rectal cancer from two institutions. Electric pulses were delivered employing the EndoVE^®^ device (Mirai Medical, Galway, Ireland), a single-use electrode coupled to a standard endoscope. The device’s active part is a chamber of approximately 2.5 cm^3^ containing two plate electrodes and is connected to a vacuum system to draw the tumour tissue into the chamber. The device allowed coverage of the tumour surface ranging from 25% to 100% and a complete response in 3 of 7 patients following a second course of treatment [179]. Regarding these preliminary results, ECT may represent an option to achieve tumour debulking and control local bleeding in patients unsuitable for more invasive treatments. More recently, treatment delivery has been performed using the VG-ECT setting or the Stinger^®^ device (IGEA S.p.A., Modena, Italy), a novel endoscopic pulse applicator capable of deploying five needle electrodes from a single, 5/10-mm diameter shaft [180,181,182].

### 4.7. Research Directions—ECT

Current research directions in ECT cover different topics that partly overlap with other PEF-based therapies (Figure 5).

#### 4.7.1. Aesthetic Outcome

Skin cancer and cutaneous metastases represent the most frequent indication for ECT. This, coupled with the fact that the treatment intent is primarily palliative, makes aesthetic results a priority. Therefore, particularly in the face, clinical success is inextricably linked to patient perception. Thus, whereas the eradication of an SCC is easily assessed, aesthetic outcomes have not been assessed objectively. The few available reports indicate encouraging patient-reported outcomes, although results rely on heterogeneous patient populations and different questionnaires [183,184,185]. Future studies in this area should focus on reducing dermatologic toxicity and implementing assessment criteria that reflect patient and clinician insights into what constitutes a meaningful clinical benefit. 

#### 4.7.2. Quality of Life

Since its introduction in clinical practice, ECT has been proposed as an alternative low-invasive treatment. Several studies support its safety and tolerability, which generally translate into high patient acceptance rates and favourable QoL impact [185,186,187]. Interestingly, a recent study from the InspECT group in patients with melanoma indicates the value of baseline QoL as a predictor of response to ECT; moreover, in the same study, patient trajectories—investigated through the EuroQoL questionnaire—allowed clarification of novel critical aspects of the patient experience that need to be addressed [154].

#### 4.7.3. Exploiting Biological Factors

The interest in deciphering the molecular mechanisms that govern the ECT mechanism of action stems from the observed variability in effectiveness across histotypes [14,188]. From a clinical standpoint, patient selection relies solely on clinical factors such as tumour size, tumour (sub-)histotype, and previous oncological treatments, whereas no guiding biomarkers exist. To this aim, researchers have identified two major areas of interest, i.e., tumour cells and tumour microenvironment (vasculature, extracellular matrix, and immune infiltrate), and a roadmap of future research directions has been recently proposed [189]. Identifying reliable biomarkers of response may improve patient selection and increase ECT efficacy by customising treatment parameters, manipulating the tumour and its microenvironment, and developing therapeutic combinations.

#### 4.7.4. Combination with Immunotherapy

Monoclonal antibodies directed at PD-1, PD-L1, and CTLA-4 represent one of the most transformative advances in cancer treatment. However, the full potential of checkpoint blockade is not entirely realised, as only a minority of patients achieve a durable response and long-term survival benefit, and systemic immunotherapies can cause severe auto-immune toxicities. Of note, EP-based therapies can potentially elicit synergistic effects with these agents [102,190]. In this setting, ongoing research aims to improve local control and boost the systemic immune response by generating potent priming of antitumour immunity. However, the increasing number of checkpoint inhibitors entering the clinical practice and the heterogeneity of ECT modalities (drug, drug dose, electrodes, applied pulses, treatment indications, criteria for re-treatment) allow multiple combination strategies, thus making the individuation of their optimal schedule a non-straightforward task.

##### Melanoma

Despite some promising reports on the association of local treatment with ECT and novel systemic immunotherapy with checkpoint inhibitors in patients with melanoma [153,191,192], there is no standardised approach. In particular, the best therapeutic combination and treatment schedules remain to be established. In this regard, a phase-2 non-randomised multicentre study is enrolling melanoma patients to determine whether concomitant Pembrolizumab and ECT with intravenous bleomycin are safe and improve the local and systemic response (ClinicalTrials.gov Id: NCT03448666).

##### Hepatocellular Carcinoma

Regarding intra-abdominal malignancies, HCC lends itself to a combined treatment approach [193]. HCC is an immunologically hot tumour responsive to immune checkpoint inhibitors in clinical trials. Moreover, the rich vasculature of HCC makes it an ideal target for ECT antivascular effects. Finally, ECT has the potential to be combined with gene electrotransfer (plasmid DNA coding for IL-12) either in the same EP session (simultaneous liver injection) or through gene electrotransfer into the distant skin of muscle [193].

##### Head and Neck Squamous Cell Carcinoma

Checkpoint inhibition with pembrolizumab or nivolumab is standard-of-care in patients with metastatic or recurrent/persistent head-and-neck squamous cell carcinoma (HNSCC) not amenable to curative RT or surgery [194]. Despite being investigational, ECT has been demonstrated to be a safe and effective option in the EURECA multicentre registry study. In particular, it was associated with a 56% ORR and 54% 1-year OS [195]. On these bases, the combination of immunotherapy and checkpoint inhibitors is envisioned as a potential new approach for these patients [196].

#### 4.7.5. Evaluation of Local Response

Reliable assessment of tumour response to ECT is fundamental to patient counselling and planning of retreatment. Additionally, it has crucial implications for results reporting in clinical studies [197]. Historically, an adaptation of the Response Evaluation Criteria in Solid Tumors (RECIST) has been adopted as the most feasible and reliable tool to assess ECT effectiveness in skin cancers [198]. However, ECT produces an intense inflammatory reaction in the treated tissues, and tumour response may be challenging to assess with clinical examination or radiological imaging [172]. Depending on eventual concomitant treatments, histological verification or a watchful waiting approach can be an option in patients with superficial tumours [150]. In evaluating deep-seated malignancies, instead, perfusion and diffusion MR-derived parameters, Choi, and PERCIST criteria seem more performant than morphological MR and CT criteria [199].

#### 4.7.6. Tumour Sensitivity to Histologic Subtype

According to the most recent comprehensive analysis of the InspECT register, the response rate to ECT varies according to the tumour type [200]. Additionally, preliminary evidence indicates that, within the same histotype, there may be a significant difference according to the specific tumour subtypes. For instance, according to an Italian multicentre retrospective study on breast cancer patients, those with oestrogen-receptor-positive, low-Ki-67 tumours (the so-called “luminal A-like” breast cancer) were most likely to achieve a CR and more durable local control following ECT compared with other subtypes [201]. Similarly, among BCCs, retrospective data indicate that aggressive sub-histotypes may be less sensitive to ECT [202]. If confirmed prospectively, these observations will allow for refining patient selection and customising treatment applications.

#### 4.7.7. Oldest-Old Patients

With the ageing of the general population, alternative nonsurgical treatments such as ECT are on the rise. According to the data from the InspECT register, ECT can be applied safely and with the same efficacy in the oldest-old population as in the other age groups [203].

#### 4.7.8. Neoadjuvant ECT

With the increasing confidence in ECT techniques, novel potential applications have emerged. For example, in the neoadjuvant setting, ECT may be applied, similarly to radiotherapy, as a downstaging treatment for large tumours where the radicality of upfront surgical resection is at risk. For instance, in patients with primary vulvar squamous cell carcinoma (V-SCC), ECT may be an attractive option before surgery to reduce tumour size and the extent of surgical resection. In a monocentric experience, nine patients underwent neoadjuvant ECT; seven achieved CR/PR, and six could be managed with more conservative resection with no detriment to tumour control [204]. Intriguing data also come from another monocentric experience in 41 patients with primary or secondary malignancies of various histotypes (mostly SCC) and median tumour size of 6 cm, who were deemed at risk of incomplete resection. In this study, ECT was associated with a 55% median reduction of tumour volume, and 25 patients (61%) achieved tumour clearance after surgery, with no relevant surgical morbidity [205]. Prospective multicentre studies are needed to explore further and standardise this strategy. An ongoing phase II randomised trial in patients with locally advanced rectal cancer is going in the same direction. This study investigates ECT as an adjunct to neoadjuvant therapy to boost local response and increase the rate of patients suitable for organ-sparing surgery [206].

#### 4.7.9. Adjuvant ECT

ECT can also be applied in the adjuvant setting to sterilise the surgical bed following tumour resection. While this approach has been applied extensively in veterinary medicine, in humans, there is only a proof-of-concept case report on breast cancer so far [207].

#### 4.7.10. Bleomycin De-Escalation

According to the ESOPE guidelines, the standard dose of intravenous bleomycin is 15,000 IU/m^2^, with a lifetime cumulative dose of 400,000 IU [150]. Since most patients with skin metastases receive repetitive ECT applications, reducing the bleomycin dose may reduce the risk of toxicity and extend treatment indications to new groups of patients. Summary data from retrospective evidence indicates that the efficacy of ECT with de-escalated bleomycin is comparable to ECT with the standard dose [188]. However, due to limited numbers and heterogeneity in the degree of dose de-escalation among studies, only prospective randomised studies with a larger cohort of patients will shed light on this critical procedural aspect.

#### 4.7.11. Intra-Abdominal Malignancies

A phase I-IIb multicentre randomised trial is evaluating the adjunct of laparoscopic ECT to systemic chemotherapy (FOLFOXIRI regimen) in patients with locally advanced pancreatic cancer. In this study, electric pulses are delivered laparoscopically through a flexible, expandable needle electrode [208]. Another phase I-II study was planned to investigate the feasibility of the intraoperative application of ECT on the surgical bed following surgical resection in patients with resectable pancreatic cancer. Unfortunately, the investigation was terminated following the enrolment of three patients due to safety concerns (NCT04281290).

#### 4.7.12. Endoscopic Application

##### Oral Cavity and Oropharynx

HNSCC is associated with a risk of local failure as high as 60%. Unfortunately, treatment of recurrence is demanding for patients, owing to the involvement of delicate anatomical structures and the application of multi-drug protocols. Recently, PDL-1 has been approved for patients with PDL-1-positive tumours. However, 10–15% of recurrent/metastatic HNSCC are PDL-1 negative and cannot benefit from immunotherapy. Therefore, a phase-IIb randomised study has been launched to enrol 96 patients with PDL-1 negative HNSCC of the oral cavity/oropharynx treated with standard systemic treatment (cetuximab + platinum-based agent + 5-fluorouracil) or ECT with bleomycin and assess whether local therapy improves response compared with systemic treatment alone [209].

##### Gastrointestinal Tract

Although little is known about the safety of ECT application in the gastrointestinal tract, ECT is being actively investigated in HNSCC, oesophageal, gastric, and rectal cancer (Table 1). These malignancies represent a technically and clinically challenging field of application owing to the treatment-associated risk for immediate and delayed complications, mainly bleeding and visceral perforation. The ongoing studies include patients at different disease stages (primary resectable, primary locally advanced, recurrent, or metastatic disease) and investigate ECT administered employing different devices and in the frame of various therapeutic strategies (alone or in combination with systemic treatment). Due to the recent introduction of dedicated endoscopic electrodes, the main aim of these studies will be to confirm the feasibility and safety of the procedure [178,181].

#### 4.7.13. Research Methods

Interpreting data from studies employing dissimilar populations, treatment, and reporting methods is challenging. The necessity to improve the evidence basis of ECT, and, in particular, the need to improve the reporting of clinical studies, has been pointed out by the authors of reviews and meta-analyses [152,210]. To this aim, in 2016, the InspECT group introduced a standardised checklist including the desirable parameters (trial design, patient population, treatment details, outcome assessment, and analysis and interpretation) to assist researchers in producing high-quality clinical data and standardised reporting [197]. The next step will be to assess whether these recommendations improved the quality of reporting of clinical ECT. To this aim, we endorse multi-institutional collaboration [149], high-quality clinical databases (e.g., the InspECT registry or the Registry on Percutaneous ElectroChemoTherapy [RESPECT, NCT05267080] on patients with liver tumours led by the Cardiovascular and Interventional Radiological Society of Europe), and comparative studies with other local therapies [211,212].

## 5. Calcium Electroporation

### 5.1. Principles of Ca-EP

Calcium EP was first described as a cancer treatment option in 2012, where in vitro and in vivo data showed that EP in the presence of calcium leads to rapid tumour cell necrosis associated with acute and severe ATP loss [6]. It has been shown that calcium EP does not lead to DNA damage but instead works through ATP depletion, including mitochondrial dysfunction elicited by cellular calcium overload [213]. Calcium EP is reviewed in more detail elsewhere [214]. Calcium has some clear advantages: it has an extremely long shelf-life, is in production already, and is inexpensive. The non-mutagenic nature of the molecule also means it can be used to treat benign or premalignant conditions, and working with calcium does not require any precautions (as does chemotherapy). However, calcium has to be injected locally to avoid systemic hypercalcemia, and thus in some instances, electrochemotherapy using intravenous bleomycin may be an advantage. When performing in vitro studies with calcium EP, a dose of one mM is used, and indeed, when working in vitro, each cell will be exposed to a larger amount of calcium in the suspension.

On the contrary, in vivo and in patients, doses of up to 225 mM are used since the extracellular volume is small, calcium is absorbed to, e.g., proteins, and there is a loss to the systemic circulation. In preclinical and clinical studies, tumour tissue is more sensitive than normal cells. This phenomenon is not seen in cell suspension studies, but it is straightforward when examining spheroids [215] or tumours [23]. The role of PMCA (plasma membrane calcium ATPase), a molecule that expels calcium from the cell interior, has been observed, indicating that a higher ability of normal cells to get rid of extracellular calcium might be important [23].

### 5.2. Clinical Studies

After the intriguing preclinical results, it was decided to start a clinical trial. In order to immediately know how calcium EP would compare to electrochemotherapy, this was designed as a double-blinded randomised trial [216]. Thus, patients with cutaneous metastases agreed to be treated with local injection of either calcium chloride (220 mM) or bleomycin (1000 IU/mL), followed by standard EP protocol with trains of 8 pulses using 0.1 ms at 400 V. Six patients with cutaneous metastases from breast cancer and one patient with disseminated malignant melanoma were included. A CR for calcium EP of 66% was found vs. 68% in the ECT-treated metastases. A trial following the same protocol included six melanoma patients and one patient with metastases from breast cancer, finding lower response rates, yet both options were efficacious [15]. Interestingly, one patient with malignant melanoma had responses in both treated and untreated lesions, indicating a systemic immune response (despite no other treatment being administered) [217]. Preclinical studies further corroborate the induction of systemic immune response, e.g., showing memory against reinduction of the tumour when previously treated with calcium EP [218]. Currently, data on the treatment of cutaneous metastases have sustained response rates with very few side effects [15,216,217,219]. A multicentre study is looking to investigate response rates in a larger and more heterogenous patient cohort [220]. Interestingly, a recent study on qualitative interviews shows improved QoL after treatment of cutaneous metastases [221]. A case report on lymphoma treatment has also shown promising results [222]. A first study on head and neck cancer mucosal tumours has shown objective response in three of six treated patients, with one patient achieving a durable CR [223]. The first clinical studies on oesophageal cancer [224] and colorectal cancer [225] using an endoscopic electrode to apply calcium EP have shown that this treatment is safe and that tumour reduction can be achieved. In gynaecological cancer, calcium EP has been used for cutaneous and internal metastases from ovarian cancer and vulvar cancer, as well as for vulvar dysplasia [226]. As calcium is non-mutagenic, there is also a prospect for treating benign conditions, e.g., keloid scars, which have also been investigated [227].

### 5.3. Research Directions—Calcium EP

Current research trends fall into three categories. First, calcium interacts differently with the membrane than other molecules and has a different mechanism of action when exerting cell death in malignant cells. Further illumination of this will enable further sophistication of the method. Second, initial clinical experiences have been small trials with safety as the primary endpoint. Therefore, investigating larger populations and different histologies will be the next step. Third, the immunostimulatory properties of calcium EP that can lead to abscopal effects will be further investigated and exploited.

## 6. Tumour-Treating Fields

### 6.1. Principles of TTFs

TTFs employ low-intensity (1–3 V/cm), intermediate-frequency (100–300 kHz), alternating electric fields to disrupt the mitotic spindle, leading to chromosome missegregation and apoptosis. In situations where biological processes require precise spatial and temporal alignment, such as mitosis, externally applied electric fields can disrupt this process [228]. TTFs frequency is tuned to specific cancer cell types [10]. The antimitotic effect of TTF therapy was demonstrated in multiple cell lines using different frequencies. Furthermore, TTFs also affect dividing cells through the dielectrophoretic effect and structural disruption associated with membrane blebbing [228]. More recently, other mechanisms of action have been elucidated, including biophysical (decreased cell migration, increased permeability of the cell membrane and blood–brain barrier) and biological (autophagy, replication stress, immune activation) effects [10,229]. Still, their mechanisms of action are not entirely understood. TTFs received FDA regulatory approval for newly diagnosed or recurrent glioblastoma (200 kHz) and pleural mesothelioma (150 kHz) based on the results of the EF-11, EF-14, and STELLAR trials [16,230,231]. Of note, in the EF-11 and EF-14 randomised phase-3 trials, the adjunct of TTFs to standard-of-care treatment was associated with a significant increase in OS compared with standard treatment.

Furthermore, TTFs are under active clinical investigation to treat thoracic and abdominal cavity malignancies, such as non-small cell lung cancer (NSCLC), brain metastases from NSCLC, pancreatic cancer, ovarian cancer, hepatocellular and gastric adenocarcinoma [232]. From a practical standpoint, TTF is the only genuinely non-invasive PEF-based therapy because pulse transducers are applied as wearable arrays on the skin surface. As such, the risk of systemic toxicity is low, as well as the risk of additive toxicity with systemic treatment; on the other hand, local toxicity is limited to mild-to-moderate dermatologic side effects (i.e., contact dermatitis, pruritus, hyperhidrosis, pressure necrosis, and skin erosions where the skin is in contact with the adhesive or hydrogel of the pulse applicator). Overall, the incidence of any-grade dermatologic toxicity ranges from 16 to 90% in the major studies, with grade-3 toxicity ranging from 0 to 18% [16,231,233,234,235,236,237]. These can be treated with topical agents and generally do not affect patient compliance.

Nevertheless, dedicated management guidelines have been recently developed for patients treated with TTFs on the scalp and in the thoracic region due to their relevant implications for patient QoL, adherence to the treatment schedule, and effective application of skin transducers [238,239]. As a safety note, compared with other PEF-based approaches using needle electrodes instead [240], TTF does not carry any risk of tumour seeding or bleeding. Importantly, as in IRE and VG-ECT, evidence suggests that results could be improved by precisely defining TTF dosimetry, implementation of rigorous treatment planning (e.g., through patient-specific computational models) and accurate delivery of TTF to the tumour bed [241,242,243,244].

### 6.2. Glioblastoma

Glioblastoma (GBM) is the most common and lethal brain tumour. Low immunogenicity, complex genetic makeup, and the blood–brain barrier represent a hostile environment to antigen exposure, immune cell infiltration and drug penetration. Beyond an antimitotic effect, TTF can target cancer cells through several mechanisms of action (e.g., decreased cell migration, increased cell membrane permeability, impaired angiogenesis, DNA damage, immunogenic cell death, innate immunity, pyroptosis—a highly inflammatory form of programmed cell death—and release of proinflammatory cytokines and chemokines), which makes TTF an attractive modality to include in combinatorial approaches [245]. The most intriguing emerging evidence suggests that TTFs induce inflammation and stimulate antitumour immunity. For instance, Chen et al. demonstrated that TTFs produce proinflammatory cytokines and type-1 interferons; additionally, in syngeneic murine GBM models, TTF-treated GBM cells induced antitumour memory immunity, which translated into a 42–66% cure rate. Finally, using single-cell and bulk RNA sequencing of peripheral blood mononuclear cells in GBM patients, they detected robust post-TTF T-cell activation and clonal expansion [229].

Collectively, these results place TTFs in the unique dual position of local inflammasome activators without the systemic side effects of drugs and, in perspective, a tumour-specific immunising platform. Regarding clinical benefit, TTFs were first investigated in 237 patients with recurrent GBM (TTF at 200 kHz vs. standard salvage chemotherapy). Unfortunately, despite a more favourable statistical trend on oncological outcomes, overall response rate (ORR) and overall survival (OS) were not statistically significant (14% vs. 9.6%; 6.6 vs. 6.0 months, respectively). However, a safer toxicity profile and better QoL outcomes favoured TTF treatment compared with systemic chemotherapy [230]. Following this study, a multicentre randomised trial enrolled 695 GBM patients whose tumour was resected or biopsied and had completed concomitant radiochemotherapy. Patients were randomised to TTF plus maintenance temozolomide or temozolomide alone. TTF therapy consisted of low-intensity, 200 kHz frequency, alternating electric fields and was delivered through four transducers applied on the shaved scalp and connected to a portable device over 18 h per day. Given the statistically significant improvement in survival in the experimental arm (median OS, 20.9 vs. 16.0 months; HR 0.63, 95% CI, 0.53–0.76; *p* < 0.001), the Food and Drug Administration (FDA) approved TTF treatment in 2011. Of note, a persistent benefit was observed even after TTF suspension. Additionally, the patients who developed disease progression could continue TTF in association with second-line chemotherapy and also, in this setting, TTFs were associated with longer OS (11.8 vs. 9.2 months, *p* = 0.049) [16].

### 6.3. Mesothelioma

Pleural mesothelioma is an aggressive tumour with a poor prognosis. Standard chemotherapy regimens include platinum agents and pemetrexed; recent evidence suggests that immunotherapy has a role in a subset of patients. The FDA approved TTF for mesothelioma based on the single-arm multicentre phase II STELLAR trial, in which patients with unresectable disease were treated with TTF at 150 kHz frequency and concomitant pemetrexed plus cisplatin/carboplatin. Among the 80 enrolled patients, ORR was 40%, median PFS 7.6 months, and median OS 18.2 months. Skin reaction was the only adverse event associated with TTF and was graded as 1–2 in 66% of patients and grade 3 in 5% of patients [231]. However, one of the unresolved issues concerns the different sensitivities of mesothelioma histological subtypes, where patients with biphasic and sarcomatoid histotypes have the shortest PFS and OS. Understanding the underlying causes of this behaviour may allow the design of a more efficacious treatment schedule. In a preclinical model, Mannarino et al. provided convincing evidence that TTFs induce specific effects on cell proliferation in different subsets of mesothelioma cells and provided a mechanistic rationale for future combination therapies [246].

### 6.4. Lung Cancer

TTFs were investigated in the advanced or metastatic setting in non-small cell lung cancer. In a phase I-II study with 41 patients with stage IIIB-IV disease, adding TTFs to second-line pemetrexed conferred better disease control and a survival benefit than pemetrexed alone. Median OS was 13.8 months, without treatment-related severe adverse events [237].

### 6.5. Pancreatic Cancer

The phase II PANOVA trial evaluated the combination of TTF (18 h/day) and gemcitabine with or without nab-paclitaxel in 40 patients with newly diagnosed locally advanced or metastatic pancreatic cancer. The combination was safe (grade-3 dermatologic toxicity was 17%, resolved with a temporary reduction of TTF use) and tolerable (patient compliance, 68–78%). Additionally, PFS and OS compared well with historical data from other trials [234]. The ongoing phase 3 PANOVA-3 trial (NCT03377491) aims to confirm these results on a larger cohort.

### 6.6. Ovarian Cancer

Similar to pancreatic cancer, also in ovarian cancer, the phase II single-arm INNOVATE trial confirmed the safety of TTF in combination with paclitaxel in 31 patients with recurrent, platinum-resistant ovarian carcinoma. The median PFS was 8.9 months, and the median OS was not reached. These preliminary findings need verification in the phase III INNOVATE-3 trial (NCT03940196) [247].

### 6.7. Research Directions—TTF

#### 6.7.1. Preclinical Research

Regarding TTF mechanisms of action, emerging evidence from studies conducted in vitro and in vivo models supports their role in stimulating antitumoural immunity [229,248]. Another line of research investigates the effect of TTFs on cellular migration. In this regard, it has been shown that there is a potential reversion of the epithelial-mesenchymal transition, which represents a critical step in the metastatic spread [249,250]. Recently, it has also been discovered that TTFs have a role in cellular membrane permeability by increasing the number and the size of membrane pores, which could enhance permeability [251]. These findings are exciting when developing therapeutic strategies to overcome the blood–brain barrier [252]. Preclinical studies have also been conducted to evaluate the in vitro response of various cancer cells and TTF combined with medical drugs (for example, liver cancer cells and sorafenib, where HCC cells showed a response to TTF, and this response was associated with an improvement in the sensitivity of HCC to sorafenib [253]. A comprehensive overview of emerging TTF mechanisms of action is provided by Rominiyi et al. [11]. Regarding treatment delivery, TTF dosimetry and treatment planning are still in their infancy. In this regard, Bozon et al. have provided an overview and research roadmap. In summary, crucial points will understand how dose distributions influence tumour progression patterns, model the electric properties of tumour tissues and surrounding structures, and identify advanced imaging techniques to monitor tumour response [254].

#### 6.7.2. Clinical Research

In the clinic, TTFs are being evaluated for various malignancies. In lung cancer, the LUNAR study (NCT02973789) has been launched to evaluate the effect of TTF on the standard of care after platinum failure. Various studies explore the role of TTF combined with chemotherapy (gemcitabine and paclitaxel, NCT05653453) or stereotactic body radiation (NCT05679674) in locally advanced pancreatic cancer. Other studies evaluate the use of TTF in treating metastases, such as brain metastases from small-cell lung cancer (NCT03995667). Finally, an exciting field of research is the combination of TTF and immunotherapy, which could potentially create novel synergic approaches, especially in glioblastoma [229,252]. TTF plus pembrolizumab and temozolomide for newly diagnosed glioblastoma showed improved median PFS (11.2 months, with 24% of patients having a complete or partial response) versus historical control data on TTF-temozolomide alone, according to the results from the phase 2 2-THE-TOP trial (NCT03405792).

## 7. Discussion

Clinicians are becoming increasingly aware that PEF-based therapies are opening new windows of opportunity for patients with cancer. Evidence shows that IRE/H-FIRE, GET, ECT, Ca-EP, and TTF offer tangible benefits. The recent WCE in Copenhagen represented a unique opportunity to gather leading experts and up-to-date results with PEF-based therapies, an emerging composite platform providing effective therapeutic options to the broader cancer population. Despite being a niche, PEF-based therapies are now integral to the portfolio of multidisciplinary cancer teams dealing with solid tumours. This review provides an up-to-date, agile overview and aims to address the fragmentation in this field produced by different maturation, equipment, and regulatory approval. As illustrated, these approaches take advantage of electric fields, variably gauged to produce the intended biological effect, to exert a perturbation of tumour cell structures, e.g., on the cytoplasmatic membrane or the cytoskeleton. Of note, they can target most solid tumours with encouraging results regarding safety, efficacy, and patient-reported outcomes [154,221].

However, PEF-based therapies remain at different stages of development and have inconsistently received regulatory approval worldwide. Different reasons may explain the status quo, including safety and economic concerns and the initial lack of supporting evidence. Although beyond the scope of this review, it is worth noting that there is increasing evidence of the safety and tolerability of these therapies, particularly when treating superficial tumours with GET, standard ECT, or Ca-EP [15,24,185,255]. Furthermore, TTFs are well tolerated by glioblastoma patients, who generally report only mild dermatologic side effects and have shown high compliance rates [230]. Finally, even IRE/H-FIRE and VG-ECT are associated with low complication rates, like other consolidated percutaneous therapies [80,89,177].

Regarding costs, PEF-based therapies are generally perceived as a valuable option. Still, it is impossible to make general considerations because there are only sporadic studies, mainly focusing on single approaches [256,257,258,259], and a lack of comparative analyses. Finally, the best available evidence for PEF-based therapies is still heterogeneous, ranging from prospective randomised trials [16,112,230,260] and large registry-based studies [153,158,200] to small case series or case reports. In addition, these therapies have a vast range of applications and are primarily used in combination or as part of sequential multimodal strategies, thus making it challenging to conduct informative comparative trials.

Interestingly, these techniques have an increasing role in oncology and offer several advantages over surgical resection and systemic treatment: most notably, lower morbidity, increased tissue preservation, reduced costs, and short hospital stay, not to mention the opportunity to expand therapeutic indications and include patients who would otherwise not be candidates for any treatment. However, from a practical standpoint, clinicians should be aware that these procedures differ regarding the type of anaesthesia, the intended target lesion(s), and the specific skills or facilities required (Table 2). On this note, for example, IRE and VG-ECT, when applied percutaneously, invariably require an interventional radiologist’s assistance, whereas, during open surgical procedures, CT or US is critical to ensuring precise pulse delivery and treatment verification [261,262]. Similarly, TTFs share a conceptual framework for ionizing radiations (i.e., dose–response correlation, quantification of dose distribution to the target volume). As such, radiation oncologists are well-positioned to integrate TTFs into their clinical workflow [242].

Thanks to their progressive acceptance, particularly in interventional radiology [70], endoscopy [178,179,224,225], laparoscopic surgery [193], and radiation oncology [241,242], almost every tumour in the human body can now be safely targeted with PEF-based therapies. Given the vastity of the topic, the reader should be aware that the present report is not a systematic review but rather a snapshot of current clinical applications and research directions. As such, we remind that each PEF-based approach has been described in more detail elsewhere [14,78,214,263,264,265], and systematic reviews on their efficacy are also available [78,80,83,152,162,177,188,210,214,255].

Now, the current challenges pertain to refining treatment indications, individuating the most advantageous timing of treatment application, modulation of treatment intensity (e.g., pulse parameters, number of applications, drug doses, treatment frequency), selection of the target lesions (e.g., primary tumour or distant metastases), and development of combined strategies with local therapies or systemic treatment such as checkpoint inhibitors [248,266]. These issues represent a matter of debate at MDT meetings, where the treatment decision-making still lacks essential information. Nevertheless, waiting for more robust evidence, the EP community is building consensus on some critical areas (e.g., procedural aspects, dosimetry, treatment planning) to lay the basis for future studies and integrate different approaches [197,212,254,267]. Then, the next step will be addressing specific clinical questions on individual tumour types.

Moving forward, the WCE-2022 allowed the identification of a wide range of cross-sectional research directions, encompassing improving patient selection, enhancing treatment delivery, increasing treatment precision and safety, developing combined therapeutic strategies, clarifying treatment timing, refining tumour response assessment, developing patient-centred outcomes, and improving the reporting of clinical trials (Figure 5).

Due to the ancillary role played by locoregional therapies in the multimodal therapeutic strategy, patient selection is paramount to homogenise study populations and assess treatment efficacy. In this regard, researchers should aim to adopt detailed selection criteria including not only tumour histotypes but also tumour subtypes whenever applicable (e.g., BCC and breast cancer [201,202]), previous and concomitant treatments, and tumour burden [82].

Despite the development of electric protocols and the availability of standard operating procedures for some PEF-based therapies (e.g., the ESOPE guidelines [150]), some variation remains across centres in treatment delivery. For instance, bleomycin dose and electrode application (i.e., the inclusion of a safety margin) is a matter of debate among ECT users [268], whereas the standardisation of energy delivery is a controversial aspect in IRE [267]. At the same time, improving treatment precision, particularly for deep-seated lesions, can potentially ensure patient safety and improve treatment results. In this regard, image guidance and patient-specific pretreatment planning provided by interventional radiologists and radiation oncologists are the backbones of safe PEF-based interventional applications [3,59,93,172,241,243,244,254].

Interestingly, beyond providing clinical benefits such as local control and QoL, PEF-based therapies shape patient immunity and influence tumour susceptibility to immunotherapies. Hence, the immune correlates of PEF-based therapies represent an exciting field of investigation. Indeed, the positive preclinical and early clinical findings [106,107,108,147,148,269,270,271,272] have led to the development and implementation of new treatment strategies aimed at modulating the immune response and possibly enhancing the clinical response to checkpoint inhibitors [109,111,153]. Hopefully, these advances should pull researchers working on different PEF-based therapies together to build on each other’s work.

This perspective leads to a further question: can PEF-based therapies be combined? This approach is intriguing for several reasons. First, although differences in equipment may represent a logistic barrier, still, the procedures are relatively straightforward and well-tolerated by patients. Thus, unsurprisingly, researchers are pursuing their combination. For instance, ECT and Ca-EP can be easily integrated into the same therapeutic strategy for patients with skin cancers or cutaneous metastases (e.g., melanoma), thus providing an additional opportunity to overcome tumour resistance and reduce exposure to chemotherapy [109,217].

Similarly, GET could be combined with standard ECT or Ca-EP for treating superficial tumours, whereas IRE may be associated with VG-ECT for treating parenchymal malignancies. Notably, in both these clinical scenarios, several other therapeutic options exist. For instance, T-VEC, rose Bengal, imiquimod, and IL-2 are used in melanoma [152,273], whereas transarterial embolisation, chemoembolisation or radioembolisation, RFA, and MWA are all effective for liver metastases [273]. As a result, the number of potential combinatorial approaches is growing. Moreover, novel immunotherapy agents represent an additional layer of complexity towards identifying the best therapeutic strategies. At the same time, however, this represents an exciting opportunity to elicit an effective immune response and expand the number of tumours susceptible to checkpoint inhibitors [106,107,108,109,110,111,112,115,134,153,269,270].

An aspect closely interrelated with the individuation of optimal therapeutic strategies is the timing of locoregional treatment application. In this regard, PEF-based therapies may be applied, similarly to radiotherapy, with a neoadjuvant or adjuvant intent. A novel setting to explore these approaches is the treatment of unresectable/borderline resectable skin cancers or soft tissue tumours. Specifically, ECT/Ca-EP could be applied preoperatively to downstage the tumours and allow for subsequent radical resection [205] or, alternatively, following surgery to ensure sterilisation of residual microscopic disease. Of note, the same concept is being explored with IRE in treating pancreatic cancer (i.e., MA-IRE) [12,62,74].

Next, tumour response assessment is critical to assess treatment efficacy and inform patient management (e.g., the need for additional sessions or other therapies). Of note, traditional evaluation methods (clinical assessment, radiological size-based criteria) have limitations in assessing superficial and parenchymal tumours. As a result, alternative techniques and parameters are being investigated to evaluate real-time tumour coverage with electric fields and to assess response [64,199,254,261].

Historically, the evaluation of PEF-based therapies relied on tumour response. Today, patient-reported outcomes (e.g., aesthetics, QoL), along with economic evaluations, are emerging as co-primary endpoints. In addition, the current therapeutic landscape is becoming progressively crowded with treatment options, so patient preferences and financial costs are crucial to global evaluations and should be prioritised in future research.

Finally, it is expected that researchers and clinicians aim to conduct well-designed collaborative studies within each tumour pathway and adopt standardised and comprehensive methods of reporting, such as those proposed for ECT [197]. This approach will provide more information on the actual treatment applied, more reliable patient outcomes, and ultimately, may allow for performing comparative trials.

The dialogue between PEF-based therapy experts must continue to share knowledge and experiences. At the same time, the EP community needs to promote a dialogue with the broader oncology community to make further development and integration of PEF-based therapies a reality for patients. In just a few decades, PEFs have translated from the bench to the bedside and entered the oncology array of locoregional therapies. However, as summarised in Figure 5, much work still needs to be done to elucidate their role and consolidate the benefit provided to patients. Many questions remain. Future research directions lay at the crossroad of physics, biology, and oncology and must focus on preclinical, engineering, and clinical translational investigations.

## 8. Conclusions

Despite significant advances in prevention and care, patients with cancer still experience the unfortunate conditions of multifocal, locally advanced, or metastatic disease, which often is not amenable to radical surgical treatment. PEF-based therapies encompass diverse, highly effective locoregional therapies, which open new avenues. As such, IRE/H-FIRE, GET, ECT, Ca-EP, and TTF harness PEF—alone or combined with genetic material, chemotherapy, or calcium—to elicit a wide range of therapeutic effects on cancer cells. The most up-to-date results portend significant clinical benefits, including local control, QoL, and, in some cases, even survival. Conversely, these therapies are positioned at different stages of development, and surprisingly, their regulatory approval varies across countries. The procedures also rely on different equipment (pulse generators, electrodes) and competencies (image guidance, interventional radiology, endoscopy).

Nevertheless, we endorse a renewed effort by EP research groups, the oncology community, and the stakeholders to overcome these barriers, break knowledge silos, and promote collaboration. PEF-based therapies offer some distinct advantages, such as minimal invasiveness, selectivity towards tumour cells, antivascular effects (or, in the case of IRE and VG-ECT, a large-vessel sparing effect), and stimulation of the immune response. Looking forward to the 2024 WCE in Rome, it is expected that PEF-based therapies could be combined in different ways: (a) among them, (b) with other local therapies, and (c) with systemic treatment. Hopefully, combinatorial approaches may expand therapeutic options for patients, as already evident for those with pancreatic cancer and melanoma. To this aim, translational research approaches will be critical for understanding the underlying biological factors and developing rational strategies. Excitingly, several clinical investigations are underway, and progressive standardisation of the techniques, improvement in reporting, and multi-institutional collaboration will foster further advancement.

## Figures and Tables

**Figure 1 cancers-15-03340-f001:**
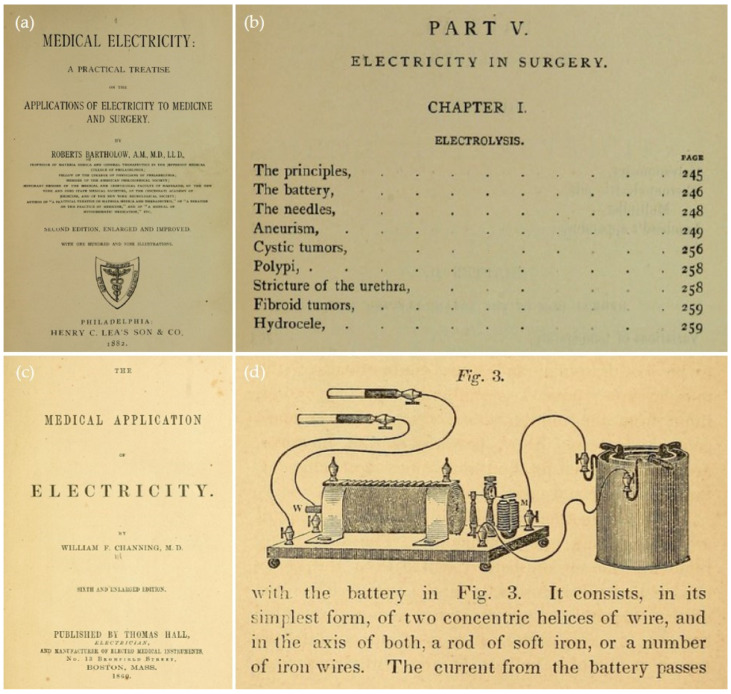
Medical application of electricity in the 19th century. (**a**,**b**) Bartholow Roberts, Medical electricity: a practical treatise on the applications of electricity to medicine and surgery. Harvey Cushing/John Hay Whitney Medical Library, Philadelphia: Lea, 1882. Available at (public domain): https://wellcomecollection.org/works/pxt787mh (accessed on 14 February 2023). (**c**,**d**) Channing, William F. The medical application of electricity. Harvey Cushing/John Hay Whitney Medical Library, Boston: Thomas Hall, 1860. Available at (public domain): https://wellcomecollection.org/works/szuexvz8 (accessed on 14 February 2023).

**Figure 2 cancers-15-03340-f002:**
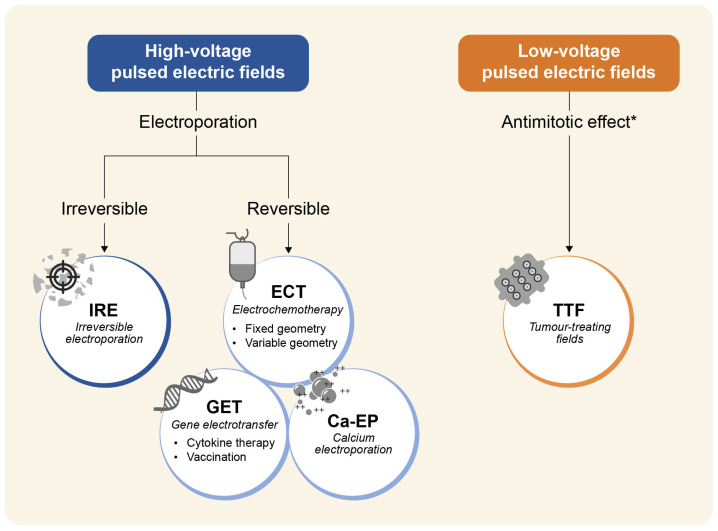
Pulsed electric fields in oncology. Classification and mechanism of action. * Other mechanisms of action of low-voltage, intermediate-frequency (100–500 kHz) electric fields include impairment of DNA repair, autophagy, stimulation of antitumour immunity (through increased expression of calreticulin, secretion of ATP and high mobility group protein 1 [HMGB1], activation of dendritic cells, production of proinflammatory cytokines, an increase in tumour infiltrating lymphocytes, DNA release, and metabolic changes in the tumour microenvironment), anti-migratory effects, and increased cell membrane permeability. Legend: HV-EF, high-voltage electric fields; LV-EF, low-voltage electric fields (1–3 V/cm); IRE, irreversible electroporation; ECT, electrochemotherapy; GET, gene electrotransfer; Ca-EP, calcium electroporation; TTF, tumour-treating field.

**Figure 3 cancers-15-03340-f003:**
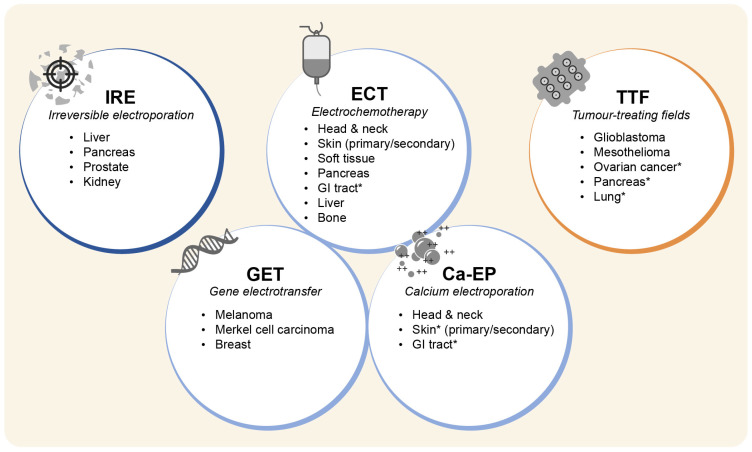
Current treatment indications of pulsed electric fields in oncology. Current GI tract indications include oesophageal and rectal cancer. * indicates investigational indications.

**Figure 4 cancers-15-03340-f004:**
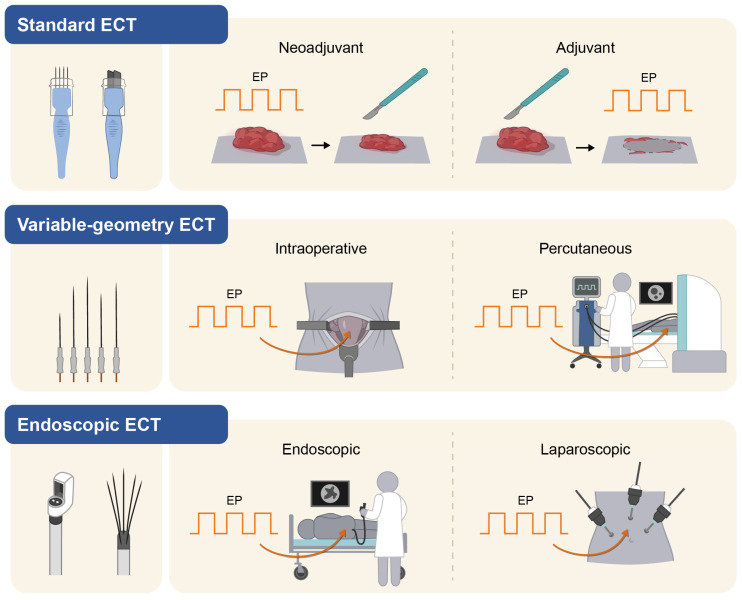
Electrochemotherapy modalities. In standard ECT, electric pulses are delivered using fixed-geometry needles or plate electrodes. In variable-geometry ECT (VG-ECT), long independent needle electrodes are inserted according to tumour geometry and size. In endoscopic ECT, electric pulses are administered using a dedicated luminal electrode connected to an endoscope or a single shaft capable of deploying an array of needle electrodes in confined anatomical spaces.

**Figure 5 cancers-15-03340-f005:**
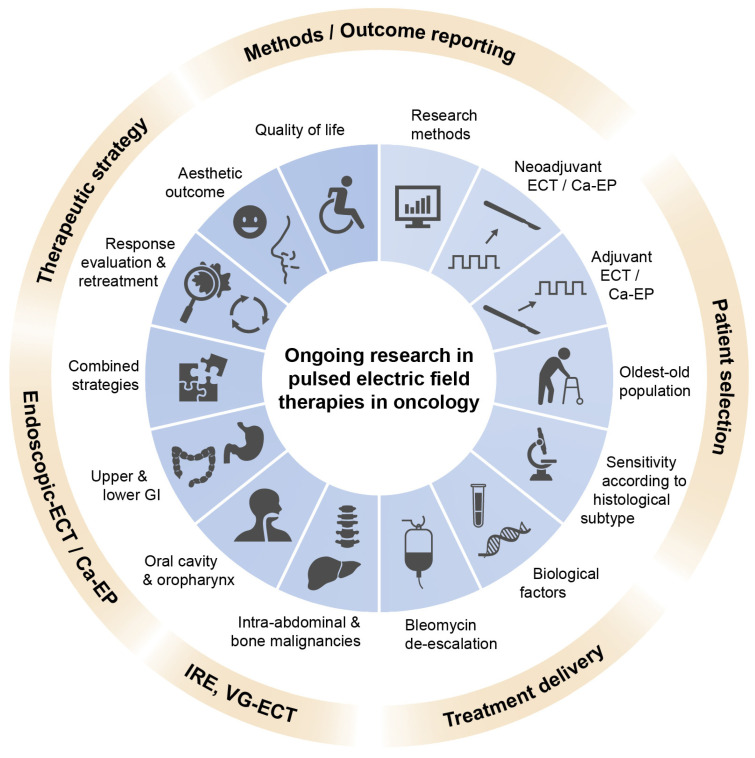
Current research directions in PEF-based therapies in oncology. The overarching and interrelated research themes include improving patient selection, enhancing treatment delivery, increasing treatment precision and safety, developing combined therapeutic strategies, clarifying best treatment timing, refining tumour response assessment, developing patient-centred outcomes, and improving the reporting of clinical studies.

**Table 1 cancers-15-03340-t001:** Ongoing clinical studies on endoscopic electrochemotherapy.

Tumour	Setting	Study Type	Treatment	Reference
HNCSCC ^1^	recurrent disease	Phase-2, randomised	ECT vs. cetuximab, CDDP/CBDCA, 5-FU	[209]
Rectal cancer	Primary, locally advanced	Phase-2, randomised	Neo-adjuvant Tx ^2^ → surgery vs. Neo-adjuvant Tx ^2^ → ECT ^3^ → surgery	[206]
Rectal cancer	Primary, locally advanced	Phase-2, randomised	Neo-adjuvant Tx ^n.s.^ → surgery vs. Neo-adjuvant Tx ^n.s.^ → ECT ^4^ → surgery	NCT03040180
Rectal/ sigmoid cancer	Primary, resectable	Phase-2, randomised	ECT → surgery vs. EP alone → surgery	NCT04816045
Gastric cancer	Inoperable	Phase-1	Bleomycin-ECT	NCT04139070

Abbreviations: CBDCA, carboplatin; CDDP, cisplatin; ECT, electrochemotherapy; EP, electroporation; 5-FU, 5-fluorouracil. ^1^ Oral cavity and oropharynx; ^2^ Long-course chemo-radiotherapy or short-course radiotherapy; ^3^ electric pulses delivered by an endoscopic needle electrode; ^4^ electric pulses delivered employing the endoscopic EndoVE electrode.

**Table 2 cancers-15-03340-t002:** Overview of PEF-based therapies.

	IRE/H-FIRE	GET	ECT	Ca-EP	TTF
**Target** **tumours**	Deep-seated	Superficial	Superficial Deep-seated	Superficial Deep-seated	Deep-seated
**Approved** **indications**	Liver mts	Melanoma	Skin	-	Glioblastoma
	Liver cancers		Head & neck	-	Mesothelioma
	Pancreas		Bone mts	-	
	Prostate				
**Investigational indications**	GI cancers	Merkel cell	GI cancers	Skin	Lung
		Breast	Liver mts	GI cancers	Pancreas
			Liver cancers	Barrett’s	Ovary
			Pancreas	Head neck	
				Keloids	
**Pulse** **application**	Needle electrodes	Needle electrode	Needle/plate electrodes	Needle/plate electrodes	Contact transducer
**Anaesthesia**	Gen	Loc/Sed	Loc/Sed/Gen	Loc/Sed	No
**Procedure** **complexity**	+++ ^1^	+++ ^2^	+ ^3^/+++ ^4^	+ ^5^	+ ^6^
**Use of CT**	no	no	yes	no	no/yes
**Approval**	USA, EU, AUS, Asia	USA	EU, AUS	No	USA, Canada, EU
**Systemic effect**	no	yes	no	Yes/?	yes
**Specific** **requirements**	Image guidance	Gene therapy licensed facility	Image guidance (only VG-ECT)	No	Patient compliance ^6^ Image guidance/radiotherapy expertise
**Costs**	++	++/+++	+/++	+	+++

^1^ General anaesthesia and image guidance are required. ^2^ Facility to prepare the genetic material. ^3^ Standard ECT on superficial tumours. ^4^ Variable-geometry ECT requires image guidance or intraoperative application; endoscopic ECT requires advanced endoscopic skills. ^5^ Ca-EP is a low-complex procedure, similar to standard ECT, with the advantage of avoiding the preparation and handling of chemotherapy drugs. ^6^ Electric pulses are delivered through external transducers applied to the skin.

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
