# Peer review of "Pulsed Electric Fields in Oncology: A Snapshot of Current Clinical Practices and Research Directions from the 4th World Congress of Electroporation"

_cancers, 2023, doi:10.3390/cancers15133340_

Round 1

Reviewer 1 Report

Campana et al wrote comprehensive review about use of pulsed electric fields and related technologies in oncology with emphasis on current clinical practices and research directions.     Manuscript is clear, relevant for the field and presented in a well-structured manner.

Paper is excellently written, understandable and comprehensive. All electroporation methods as well as fields of use and ongoing studies are listed and clearly explained. The gap in better implementation in clinical practice is recognized and underlined. References are appropriately cited, mainly most recent ones. There is no excessive number of self-citations.

Figures are appropriate and informative. They are easy to interpret and understand. Ethics statements and data availability statements are adequate.

 Similar review was not published recently. This paper is of value to the scientific community since it synthesizes the knowledge about electroporation methods while at the same time underlines practical problems with implementation into a practice. As well, paper does appropriately address problems connected to different stages of methods development, their regulatory problems as well as need for more sophisticated equipment and skilled personnel with various competencies and knowledges (i.e., image guidance, interventional radiology, endoscopy).

The statements and conclusions are drawn coherently and are supported by the listed citations.

There are only a few minor issues regarding the paper:

1. Page 12, line 393:  Authors are explaining the use of IRE in the treatment of upper gastrointestinal tract tumors. As a complication of the procedure, they have mentioned bowel perforation. This is probably mistake if they would like to address it to upper GIT. ? Esophageal ? gastric, ? duodenal, (altogether?). Please correct.

2.Page 13, line 455: please correct IL-12 into

3. Page 22, line 792: completer into complete

4. page 33, line 1197: enhancerung into enhance.

There are also a few minor spelling and font size inaccuracies.

Author Response

REV-1

Campana et al wrote comprehensive review about use of pulsed electric fields and related technologies in oncology with emphasis on current clinical practices and research directions.     Manuscript is clear, relevant for the field and presented in a well-structured manner.

Paper is excellently written, understandable and comprehensive. All electroporation methods as well as fields of use and ongoing studies are listed and clearly explained. The gap in better implementation in clinical practice is recognised and underlined. References are appropriately cited, mainly most recent ones. There is no excessive number of self-citations.

Figures are appropriate and informative. They are easy to interpret and understand. Ethics statements and data availability statements are adequate.

Similar review was not published recently. This paper is of value to the scientific community since it synthesises the knowledge about electroporation methods while at the same time underlines practical problems with implementation into a practice. As well, paper does appropriately address problems connected to different stages of methods development, their regulatory problems as well as need for more sophisticated equipment and skilled personnel with various competencies and knowledges (i.e., image guidance, interventional radiology, endoscopy).

The statements and conclusions are drawn coherently and are supported by the listed citations.

There are only a few minor issues regarding the paper:

  1. Page 12, line 393: Authors are explaining the use of IRE in the treatment of upper gastrointestinal tract tumors. As a complication of the procedure, they have mentioned bowel perforation. This is probably mistake if they would like to address it to upper GIT. ? Esophageal ? gastric, ? duodenal, (altogether?). Please correct.

R: thanks for spotting this. The text has been changed as follows: “specifically-designed catheter electrodes have been designed that are suitable for coupling with an endoscope to allow visualisation and avoid hollow viscus perforation.”

2.Page 13, line 455: please correct IL-12 into

R: Thanks for spotting this typo, which has been amended.

  1. Page 22, line 792: completer into complete

R: solved

  1. page 33, line 1197: enhancerung into enhance.

R: Thanks for also spotting this typo, which has been amended.

There are also a few minor spelling and font size inaccuracies.

R: the manuscript went through proofreading.  

Author Response

REV-2

The review presents the current status of clinical research featuring electroporation or TTFs.

The paper is well written, presenting all of the basic clinical research directions where electric

fields are employed. It should be noted that while many of the applications are covered

superficially, it’s an appropriate format considering the scope and the scale of the paper. It’s

indeed a snapshot rather than a critical review and will be useful for the electropora4on &

oncology community.

R: thanks for the words of appreciation. Indeed, this post-congress paper aims to provide an agile overview which could be informative to a wide readership. As noted during the last World Congress of Electroporation, the knowledge of these techniques is still heterogeneous, even within the electroporation community itself and clinicians. The revised version clarifies our intention to break barriers between PEF-based therapies and promote collaboration.        

Reviewer 3 Report

The manuscript is a review of pulsed electric field in oncology looking at directions from the 4th WC on electroporation.

The idea of the paper is very nice but the methodological approach is not rigorous. In the case of systematic reviews of the literature specific rules should be followed while in simple review (the present is of the last five years), the criteria for inclusion of the references in the work should be provided as well as the serched data bases. So please modify your manuscript providing methodology for your search and criteria of inclusion/exclusion of the papers as well as searched data bases and used key words.

Following the different  PEF-based technologies and the related applications are very complex and dispersive  so summary tables for clinical trials should be included at the end of each section (so for each tumor type) as well as a more comprehensive summary of the ongoign research lines should be provided in the discussion section.

Fig. 5 needs a wider description and its inclusion should be justified and well commented in the Discussion associated to this manuscript otherwise it is completely unuseless

This reviewr is also critical on the inclusion of TTF in this review. First it is not clear which type of signal they are, some time it is reported alternate field so it meas sinusoidal waves different from pulses. Other times in the manuscript TTF seems to be electric pulses so a clear description of the signal is mandatory.

The other point of concern is the fact that these fields do not induce permeabilization of cell tissue (it is unkwon their intensity on tissues) which is the common phenomenon to the PEF-based therapies. Finally the most important as TTF are the only technology among the presented applied externally (electrodes are not in direct contact with the tumor), their dosimetry is completely lack. It is not reported or even mentioned which is the electric field really experieced by the tumor. This level evidently can largely varies depending on the tumor location and dimention specific from patient to patient. So their effects seem diffcult to be homogeneous from patient to patient and a rigorous evaluation cannot be done as  electric dose is not porvided and not costant to each patient. It is like giving an unkwon chemotheraputic dose to patient which is  clearly a not scientifically sound way of research.

So, this is the main drawback of this technique and this point shoud be well adressed in the current manuscript or evidenced as a critical to this type of possible therapy. For this reason it seems better to eliminate this technology from the review as it deserves a separate and more deep discussion and presentation. The electric dose provided to each patient has to be a mandatory parameter in each well design study. Also mechanism of action are completely unclear and very badly described.

The other lacking aspect of this review is the role of the treatment planning in the the clinical practice and in the performed clinical trials, so info should be provided on exposure parameters regarding voltage and applied electric field to patients which determine the efficacy of the tretment. Also electric protocols should be mentioned for the different technologies.

It is also suggested to include H-FIRE research when IRE technology is mentioned.

This reviewer encourages the global revision of the manuscript following the criteria above mentioned and a more concise and organized presentation of the reviewed data including summrizing tables for the different technologies and clinical studies terminated and on going.

Minor issues:

caption of figure 3 need revision, as well as the figure present some asterix that need to be clarified as well as the name of cancer as GI track

A list of abbreviations is necessary.

Author Response

REV-3

The manuscript is a review of pulsed electric field in oncology looking at directions from the 4th WC on electroporation. The idea of the paper is very nice but the methodological approach is not rigorous. In the case of systematic reviews of the literature specific rules should be followed while in simple review (the present is of the last five years), the criteria for inclusion of the references in the work should be provided as well as the serched data bases. So please modify your manuscript providing methodology for your search and criteria of inclusion/exclusion of the papers as well as searched data bases and used key words.

R: Thanks for raising this issue, which allows us to clarify the aim of this manuscript (page 7, lines 179-81, Introduction, “The aim of this review…..”).

As stated in the title, this is not a review and, even less, a systematic review. Instead, it is a snapshot providing a joint report from the World Congress of Electroporation. As such, it provides updated and in-progress developments that a systematic review cannot capture. We explicitly acknowledged this intrinsic limitation in the Discussion (page 37, lines 1344-49). At the same time, we support the conduction of methodologically correct research wherever available, which has been cited throughout the paper and in the Discussion (page 37, lines 1348-49).

Following the different  PEF-based technologies and the related applications are very complex and dispersive  so summary tables for clinical trials should be included at the end of each section (so for each tumor type)

R: We agree in principle with this request but would like to remind that this is not a systematic rather a narrative review (this is now stated also in the Abstract, line 40); therefore, the creation of tables (e.g. descriptive parameters, outcome variables) would be merely subjective. For consistency with the aim of the manuscript, we would prefer not to include summary tables for each application of PEF-based therapies and let the reader choose the section of their interest. Not least, given the vast range of applications of these techniques, the number of tables would be disproportionately elevated. Finally, we refer the readership to the published systematic reviews or meta-analyses throughout the manuscript and in the Discussion (page 37, lines 1348-49)

A more comprehensive summary of the ongoign research lines should be provided in the discussion section.

R: we thank the reviewer for this request. The Discussion was implemented and now includes a more comprehensive comment on the research directions illustrated in Figure 5 (Discussion, pages 37-39). 

Fig. 5 needs a wider description and its inclusion should be justified and well commented in the Discussion associated to this manuscript otherwise it is completely unuseless

R: Fig.5 caption has been implemented and cited again in the Discussion (lines 1367 and 1449). Please see also the reply to the previous query.

This reviewr is also critical on the inclusion of TTF in this review. First it is not clear which type of signal they are, some time it is reported alternate field so it meas sinusoidal waves different from pulses. Other times in the manuscript TTF seems to be electric pulses so a clear description of the signal is mandatory.

R: We appreciate your criticism. In the revised manuscript, we provide more details on TTF (pages 31-23). Appropriate references have also been provided to the reader with more technical interests.

The other point of concern is the fact that these fields do not induce permeabilisation of cell tissue (it is unkwon their intensity on tissues) which is the common phenomenon to the PEF-based therapies.

R: Thanks for this observation. The difference in mechanism of action is now highlighted in the Introduction (page 5, lines 108-116) and clearly illustrated in Figure 2. In this regard, we find it intriguing that these approaches rely on different mechanisms of action; thus, we encourage their broader and deeper knowledge/investigation by the oncology community. Finally, as to the biological effect of this treatment modality, it is intriguing to note that TTF exert a varied range of actions beyond disrupting the mitotic spindle. These include autophagy (Kim EH, Oncogene 2019; Shteingauz, Cell Death Dis 2018), DNA damage (Kranam NK, Cell Death Dis 2017; Giladi M, Radiat Oncol 2017), immune stimulation (Diamant G, J Immunol 2021; Chen D, J Clin Invest 2022), impairment of cell migration (Siginer M, Cell Death Dis 2017; Kirson ED, Clin Exp Metastasis 2009), and also an increase of cell permeability (Chang E, Cell Death Discov 2018; Aguilar, Cancers 2021).

Finally, the most important as TTF are the only technology among the presented applied externally (electrodes are not in direct contact with the tumor), their dosimetry is completely lack. It is not reported or even mentioned which is the electric field really experieced by the tumor. This level evidently can largely varies depending on the tumor location and dimention specific from patient to patient. So their effects seem diffcult to be homogeneous from patient to patient and a rigorous evaluation cannot be done as  electric dose is not porvided and not costant to each patient. It is like giving an unkwon chemotheraputic dose to patient which is  clearly a not scientifically sound way of research.

R: Thanks for raising this crucial point.

We agree that dosimetry is crucial in oncology.  We agree that there is uncertainty in dosimetry. This is not a prerogative of TTF, however. That said, we want to highlight that dosimetry is an active field of investigation in TTF; as an example, we show here some results from the EF-14 randomised trial, showing a clear discrimination of patient survival according to the TTF dose received to the tumour bed   (Ballo, M., et al. International Journal of Radiation Oncology Biology Physics, 2019). This KM curve demonstrates patient-level dose response to TTFields. In the reported analysis, the authors showed that TTF dose could be defined as the product of the TTF average power loss density at the target region and device usage (i.e. patient compliance with the device – hours/day). Interestingly, radiation oncologists are well positioned to integrate TTF therapy into clinical practice because they already visualise 3D radiation dose distributions to and delivery to regions of interest.

(Kaplan-Meyer figure enclosed)

So, this is the main drawback of this technique and this point shoud be well adressed in the current manuscript or evidenced as a critical to this type of possible therapy. For this reason it seems better to eliminate this technology from the review as it deserves a separate and more deep discussion and presentation. The electric dose provided to each patient has to be a mandatory parameter in each well design study. Also mechanism of action are completely unclear and very badly described.

R: Thanks for this reflection. Briefly, TTF is an FDA-licensed therapy whose regulatory approval was granted following the results of well-designed randomised trials. With some of the intrinsic technical limitations raised in your question, the researchers provided reliable evidence of TTF beneficial effects in patients with glioblastoma and mesothelioma.

As to TTF dosimetry, in the revised manuscript (page 34, 1258-64), we have suggested the following reading: Tumor-Treating Fields at EMBC 2019: A Roadmap to Developing a Framework for TTFields Dosimetry and Treatment Planning by Bomzon et al. in S. N. Makarov et al. (eds.), Brain and Human Body Modeling 2020, https://doi.org/10.1007/978-3-030-45623-8_1.

On a different note, we agree that treatments should be as homogeneous as possible; in this frame, developing standard operating procedures (SOPs) is the way forward. In this regard, the European SOPs of Electrochemotherapy (ESOPE) are an excellent example of harmonisation.  

The other lacking aspect of this review is the role of the treatment planning in the the clinical practice and in the performed clinical trials, so info should be provided on exposure parameters regarding voltage and applied electric field to patients which determine the efficacy of the tretment. Also electric protocols should be mentioned for the different technologies.

R: Thanks for this request. Treatment planning has been mentioned more consistently in the manuscript: lines 140-153; 1160-64; 1258-64; 1358-60; 1381-84).

It is also suggested to include H-FIRE research when IRE technology is mentioned.

R: We thank the reviewer for this request. H-FIRE is now presented in paragraph 2.6.3 (pages 14-15)

This reviewer encourages the global revision of the manuscript following the criteria above mentioned and a more concise and organised presentation of the reviewed data including summrizing tables for the different technologies and clinical studies terminated and on going.

R: The manuscript underwent careful revision following the received suggestions, which we have appreciated and have contributed to improving the quality of the manuscript. For consistency with the aim of the paper, however, we avoided summary tables (see reply to previous questions)

Minor issues:

caption of figure 3 need revision, as well as the figure present some asterix that need to be clarified as well as the name of cancer as GI track

R: Asterisks indicate investigational indications

A list of abbreviations is necessary.

R: An abbreviation list is now provided on page 3.
